# SIMMER employs similarity algorithms to accurately identify human gut microbiome species and enzymes capable of known chemical transformations

Annamarie E Bustion[1,2], Renuka R Nayak[3†], Ayushi Agrawal[2], Peter J Turnbaugh[4,5], Katherine S Pollard[2,5,6,7,8]*

[1]Pharmaceutical Sciences and Pharmacogenomics Graduate Program, University of California, San Francisco, San Francisco, United States; [2]Institute of Data Science and Biotechnology, Gladstone Institutes, San Francisco, United States; [3]Rheumatology Division, Department of Medicine, University of California, San Francisco, San Francisco, United States; [4]Department of Microbiology & Immunology, University of California, San Francisco, San Francisco, United States; [5]Chan Zuckerberg Biohub-San Francisco, San Francisco, United States; [6]Department of Epidemiology & Biostatistics, University of California, San Francisco, San Francisco, United States; [7]Institute for Human Genetics, University of California, San Francisco, San Francisco, United States; [8]Bakar Computational Health Sciences Institute, University of California, San Francisco, San Francisco, United States

*For correspondence:
kpollard@gladstone.ucsf.edu

Present address: †Veterans Affairs Medical Center, San Francisco, United States

**Abstract** Bacteria within the gut microbiota possess the ability to metabolize a wide array of human drugs, foods, and toxins, but the responsible enzymes for these chemical events remain largely uncharacterized due to the time-consuming nature of current experimental approaches. Attempts have been made in the past to computationally predict which bacterial species and enzymes are responsible for chemical transformations in the gut environment, but with low accuracy due to minimal chemical representation and sequence similarity search schemes. Here, we present an in silico approach that employs chemical and protein Similarity algorithms that Identify MicrobioMe Enzymatic Reactions (SIMMER). We show that SIMMER accurately predicts the responsible species and enzymes for a queried reaction, unlike previous methods. We demonstrate SIMMER use cases in the context of drug metabolism by predicting previously uncharacterized enzymes for 88 drug transformations known to occur in the human gut. We validate these predictions on external datasets and provide an in vitro validation of SIMMER's predictions for metabolism of methotrexate, an anti-arthritic drug. After demonstrating its utility and accuracy, we made SIMMER available as both a command-line and web tool, with flexible input and output options for determining chemical transformations within the human gut. We present SIMMER as a computational addition to the microbiome researcher's toolbox, enabling them to make informed hypotheses before embarking on the lengthy laboratory experiments required to characterize novel bacterial enzymes that can alter human ingested compounds.

## Editor's evaluation

This paper provides important advances in utilizing chemical, metagenomic and enzyme mechanistic insights into the roles gut microbiota play in health-related chemical conversions. The authors convey results from a series of convincing studies that outline the utility of their computational

platform, one that will be useful to both specialized microbiome researchers as well as a broad audience of scientists interested in the numerous ways non-host enzymes impact host biology.

## Introduction

Humans consume a large array of foods, therapeutics, and other xenobiotics that are processed, in part, by enzymes of bacteria residing within the gut. While some bacterial enzymes are orthologous to the human metabolism repertoire, many bacteria possess metabolic capabilities distinct from our own (*Zimmermann et al., 2019a*). It is important to ascertain the extent of microbial capacity for chemical transformation because it has implications for the bioavailability, toxicity, and efficacy of the compounds humans ingest (*Koppel et al., 2017*; *Spanogiannopoulos et al., 2016*). Additionally, because the human gut microbiome differs from individual to individual, knowledge of the prevalence and abundance of bacterial enzymes must be determined before beneficial clinical and dietary decisions can be made (*Javdan et al., 2020*).

While experimental methods can be employed to expand what we know of bacterial enzymatic capabilities in the gut, the scientific community lacks genetic tools for nearly all bacterial species of the human microbiota, and heterologous expression in model organisms can fail for a plethora of reasons (*Bisanz et al., 2020*; *Patel et al., 2022*). When experimental methods are tractable, the time required is often so extended that knowledge is gained in a low-throughput manner. For these reasons, attention should turn to the employment of in silico computational methods that can guide experimentalists in their hypothesis-building process by aiding in the prioritization of substrates, species, and genes worth studying (*Aziz et al., 2018*).

Recent attempts have been made to create computational descriptions of chemical transformation by human gut bacteria, but none can be expanded to predict the metabolic capabilities of bacterial proteins with unknown function or to explore the capacity of microbial enzymes to degrade novel substrates. Two previously published methods aimed to predict known drug metabolism events within the human gut microbiome, but the accuracy of their species and enzyme predictions was limited due to the fact that both tools only consider substrates, rather than a full chemical description of substrate(s), cofactor(s), and product(s) formed in a reaction (*Guthrie et al., 2019*; *Sharma et al., 2017*). Both tools were also limited by the use of small databases that do not fully capture the diversity of the human gut microbiome (*Guthrie et al., 2019*; *Sharma et al., 2017*).

To address these gaps in accurate predictive software, we present SIMMER, a tool that combines chemoinformatics and metagenomics approaches to accurately predict bacterial species and enzymes capable of known biotransformations in the human gut. Given an input reaction, SIMMER predicts specific bacterial enzyme sequences responsible for the transformation. Each predicted sequence is annotated with taxonomy, potential function, and prevalence/abundance in human metagenomic samples.

Our key innovations are the use of full chemical representations that include cofactors employed and products produced in a reaction, the use of statistically informed sequence searches of a comprehensive human gut microbiome gene catalog, and the development of an enzyme class predictor based on reaction rather than gene sequence. As a use-case, we present evidence that SIMMER provides accurate predictions of bacterial enzymes responsible for known drug metabolism events, and we identify the likely bacterial enzyme for 88 drugs known to be metabolized by the gut microbiome for which the enzyme was previously unknown. We evaluated these predictions on external datasets, and for one reaction, methotrexate (MTX) hydrolysis, we characterized human gut microbiome bacterial metabolism in vitro across 42 strains.

## Results

### SIMMER pipeline to predict xenobiotic metabolizing enzymes

There are many desiderata for a bacterial drug metabolism predictor (*Aziz et al., 2018*). Such a tool must be able to, based on chemical analogy to known bacterial chemistry, predict bacterial species, specific enzyme sequences, and the prevalence and abundance across human samples of those predicted species and sequences. We developed SIMMER, a tool that leverages chemical and protein similarity to identify enzymes in the human microbiome that could perform a queried chemical

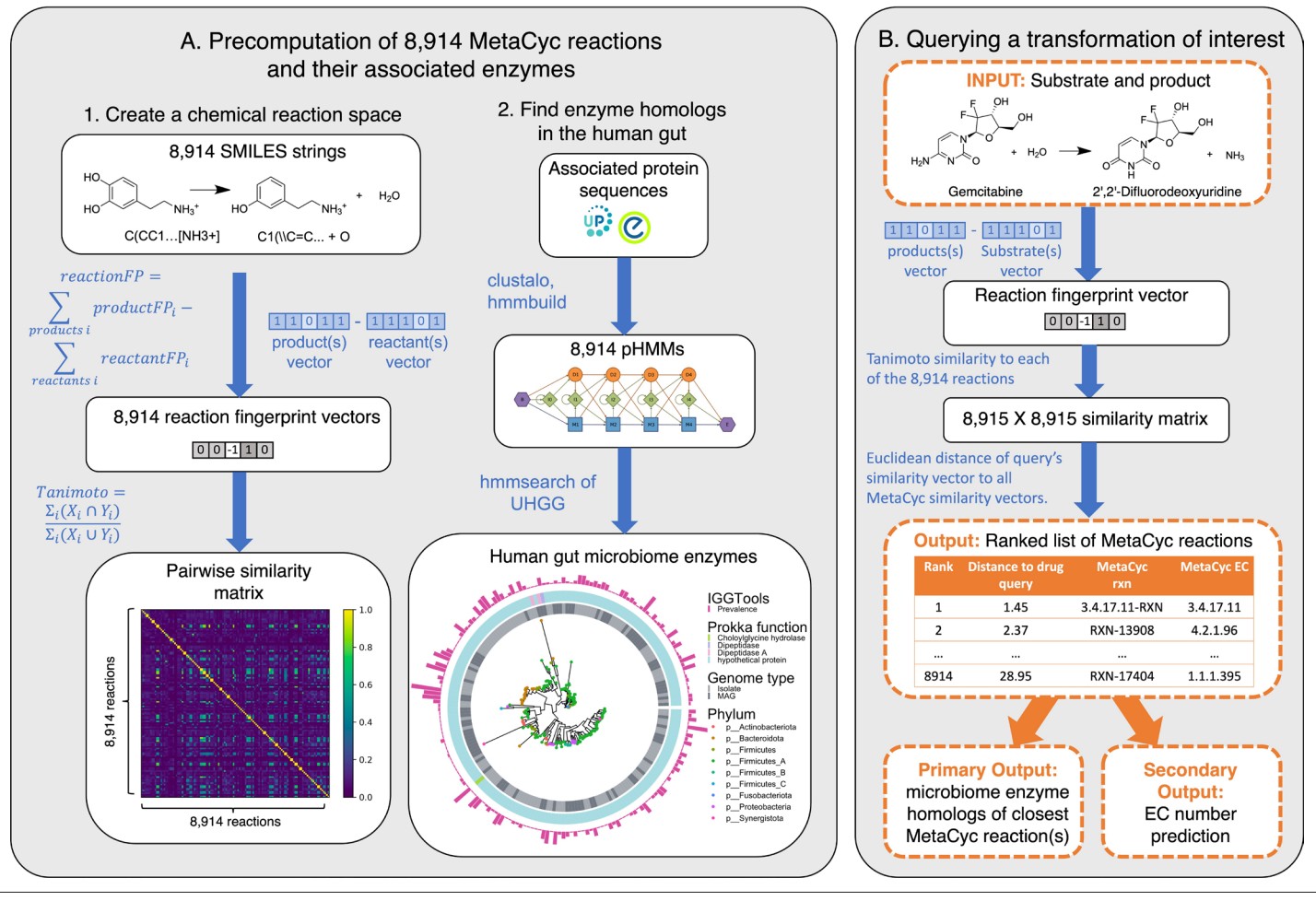

**Figure 1.** SIMMER architecture. (**A**) Precomputation on 8914 gene-annotated bacterial reactions downloaded from MetaCyc. Chemical fingerprints representing each MetaCyc reaction were created from SMILES descriptors. A latent chemical space was then created via a pairwise reaction similarity matrix based on Tanimoto coefficients. For each reaction, relevant gene sequences were retrieved from UniProt and Entrez database linkouts and used to create multiple sequence alignments and subsequent profile hidden Markov models (pHMMs) using ClustalO and HMMER3, respectively. pHMMs were used to retrieve homologs in a catalog of human gut microbiome genes. (**B**) Running a SIMMER query. After receiving a reaction query (input compound, cofactors, and products), SIMMER fingerprints the reaction and compares it to the precomputed chemical space from (**A**). MetaCyc reactions are sorted by similarity to the query. From the most similar reaction(s), human gut microbiome enzymes are reported along with their abundance and prevalence in gut microbiomes. As an auxiliary output, Enzyme Commission (EC) numbers are predicted based on enrichment in the ranked reaction list.

reaction (*Figure 1*). Given input substrate(s), metabolite(s), and any known cofactors, SIMMER predicts bacterial enzymes capable of performing the reaction and quantifies their prevalence and abundance in the human gut. SIMMER accomplishes this by chemically fingerprinting an input reaction, and then comparing it to all reactions in MetaCyc (*Caspi et al., 2020*). Enzyme annotations from the most similar MetaCyc reactions are then used as queries for a protein similarity search to find homologs in the genomes of gut bacteria. To decrease the runtime of a SIMMER query, we precomputed chemical descriptions and protein similarity searches for all reactions in MetaCyc.

SIMMER's underlying data were drawn from the MetaCyc reaction database because its small-molecule reaction descriptions each possess at least one experimentally validated enzyme annotation and because it is currently the most comprehensive database of its kind (*Altman et al., 2013*; *Caspi et al., 2020*). To build a precomputed chemical search space for SIMMER queries (*Figure 1A*), we created two-dimensional fingerprint representations for 8914 enzyme-driven reactions in MetaCyc (*Caspi et al., 2008*; *Schneider et al., 2015*). Using these fingerprints, we estimated the similarity between all pairs of reactions based on Tanimoto coefficients. To build the enzyme backbone of

SIMMER, we compiled the Uniprot and/or Entrez gene identifiers linked to each MetaCyc transformation into a profile hidden Markov model (pHMM) that represents the diversity of the enzyme family for a respective reaction. The resulting pHMMs were then used to query the Unified Human Gastrointestinal Genome (UHGG) collection of 286,997 isolate genomes and metagenome assembled genomes from the human gut environment (*Almeida et al., 2021*). Additionally, prevalence and abundance of all pHMM search hits were assessed in stool metagenomes from the PREDICT human cohort using MIDAS2, an implementation of Metagenomic Intra-Species Diversity Analysis System (MIDAS) designed for use with the UHGG catalog (*Almeida et al., 2021*; *Nayfach et al., 2016*; *Zhao et al., 2022*).

After creating SIMMER's precomputed chemical and pHMM search space, we next made SIMMER queryable (*Figure 1B*). When queried with a chemical transformation, SIMMER computes the chemical similarity of the input to all precomputed MetaCyc reactions, and sorts all MetaCyc reaction fingerprints according to their ascending Euclidean distance from the query. From this sorted list, SIMMER outputs enzymes (i.e., the precomputed pHMM search hits for the closest reactions) responsible for the query reaction. As an auxiliary function to enzyme sequence prediction, SIMMER additionally predicts an Enzyme Commission (EC) code that describes the chemical nature of the query. EC codes are four-digit identifiers of enzyme-driven reactions, where each digit describes the reaction with increasing chemical granularity (*McDonald et al., 2009*). The first digit, the EC class, describes

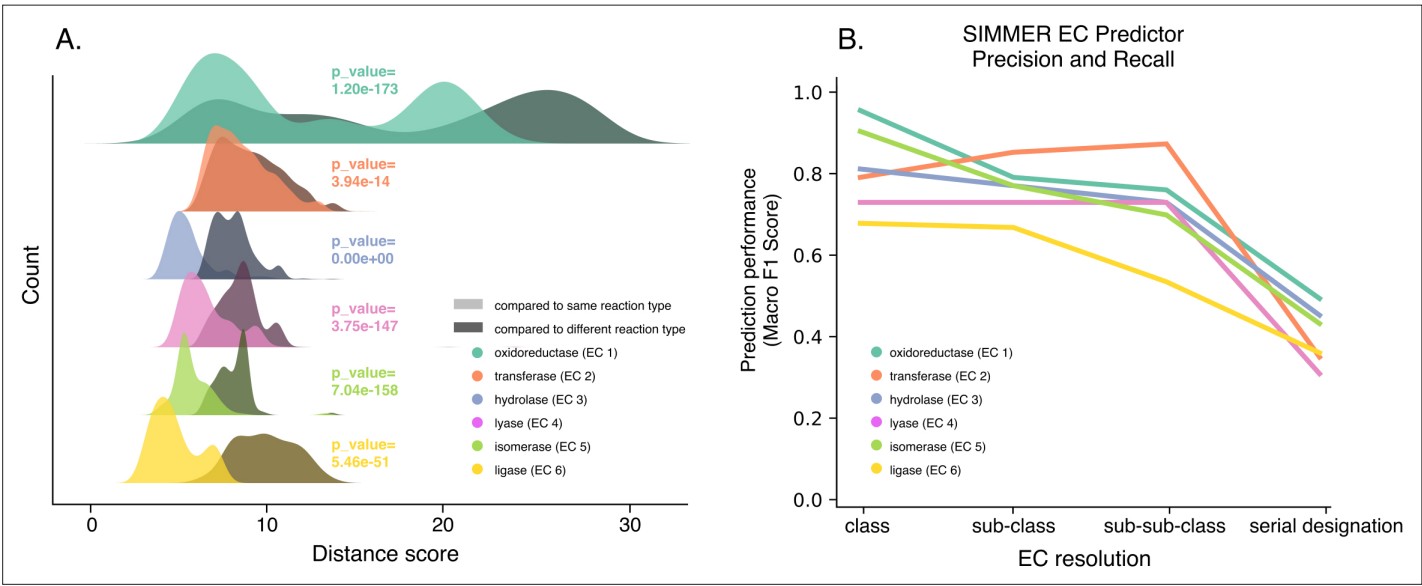

**Figure 2.** SIMMER's chemical representations capture information relevant to enzymatic reactions. (**A**) SIMMER clusters similar reactions together in chemical space. To analyze SIMMER's ability to group chemically similar reactions, we examined reaction similarity within versus without Enzyme Commission (EC) classes using the precomputed MetaCyc reaction dataset ($N = 8914$ reactions). A silhouette-like euclidean distance score was created by determining for each reaction its euclidean distance to all reactions within its EC class versus outside its EC class. For all EC classes, scores were smaller within versus without EC classes using SIMMER's chemical representation, indicating that SIMMER can detect reaction similarity within EC classes. From the pairs of distributions, we computed a Kolmogorov statistic to determine if the distributions significantly ($p < 0.05$) differed. (**B**) The $F1$-score, or harmonic mean of SIMMER's precision and recall, when predicting EC numbers on a subset of the MetaCyc database ($N = 576$ reactions total; 96 per EC class). The score is high for EC classes, and it generally decreases as an EC number's resolution increases.

The online version of this article includes the following source data and figure supplement(s) for figure 2:

**Source data 1.** Euclidean distances for reactions in the same versus other EC classes across reaction representations, and performance statistics for SIMMER's EC class predictor.

**Figure supplement 1.** SIMMER predicts an Enzyme Commission (EC) code (i.e., reaction type) for a query reaction if there is an enrichment of a particular EC at the top of the reaction list.

**Figure supplement 2.** Confusion matrices for SIMMER Enzyme Commission (EC) class predictions.

**Figure supplement 3.** Euclidean distance distributions and silhouette scores for top-level Enzyme Commission (EC) codes are resilient to fingerprint type.

**Figure supplement 4.** Euclidean distance distributions and silhouette scores for top-level Enzyme Commission (EC) codes are sensitive to chemical representation type.

broad chemistry such as whether the reaction is an oxidoreduction, hydrolysis, etc. event. The second and third digits, the EC sub-class and sub-sub-class, describe more detailed chemical information such as electron donor or transfer group identity. The fourth and final digit, the EC serial designation, often describes a reaction's specific substrate(s). We implemented and validated a novel method to predict EC codes by extending a common approach to gene set enrichment analysis (GSEA) (*Figure 2— figure supplement 1*; *Subramanian et al., 2005*). With this enrichment method, SIMMER predicted reaction types for queries with high recall and accuracy for EC classes, sub-classes, and sub-sub-classes (*Figure 2B*, *Figure 2—source data 1*, *Figure 2—figure supplement 1*, *Figure 2—figure supplement 2*).

We hypothesized that SIMMER accurately predicted the bacterial players responsible for chemical transformations due to the tool's use of a full reaction that includes reactants, cofactors, and products, rather than just substrates. We assessed this hypothesis by demonstrating that SIMMER groups similar reactions together in chemical space. MetaCyc reactions possess EC annotations that describe the chemical class of a reaction (e.g., oxidoreduction, hydrolysis, intramolecular rearrangement, etc). We queried SIMMER with all EC annotated MetaCyc reactions and demonstrated that queries group significantly closer to other reactions within their EC class than they do to reactions of a different class (*Figure 2A*). We determined that SIMMER's ability to group similar reactions in chemical space is resilient to different fingerprinting methods (*Figure 2—figure supplement 3*), but not to loss of products created and cofactors consumed in a reaction (*Figure 2—figure supplement 4*). Thus, we showed that similar reactions only cluster together in chemical space when a full reaction description (i.e., SIMMER's representation method) is employed.

Because SIMMER was created with the assumption that chemically similar reactions are mediated by sequence similar enzymes, we next ensured that similarity within SIMMER's chemical space could be used to find shared, responsible enzymes. For all MetaCyc enzymes associated with multiple reactions, one reaction was used as a SIMMER query, and the second reaction searched for in the ordered reaction list output. As a negative control, these reaction similarity results were then compared to all possible pairwise combinations of reactions not conducted by the same enzyme. SIMMER predicted high similarity between reactions conducted by a shared enzyme, and low similarity for those reactions

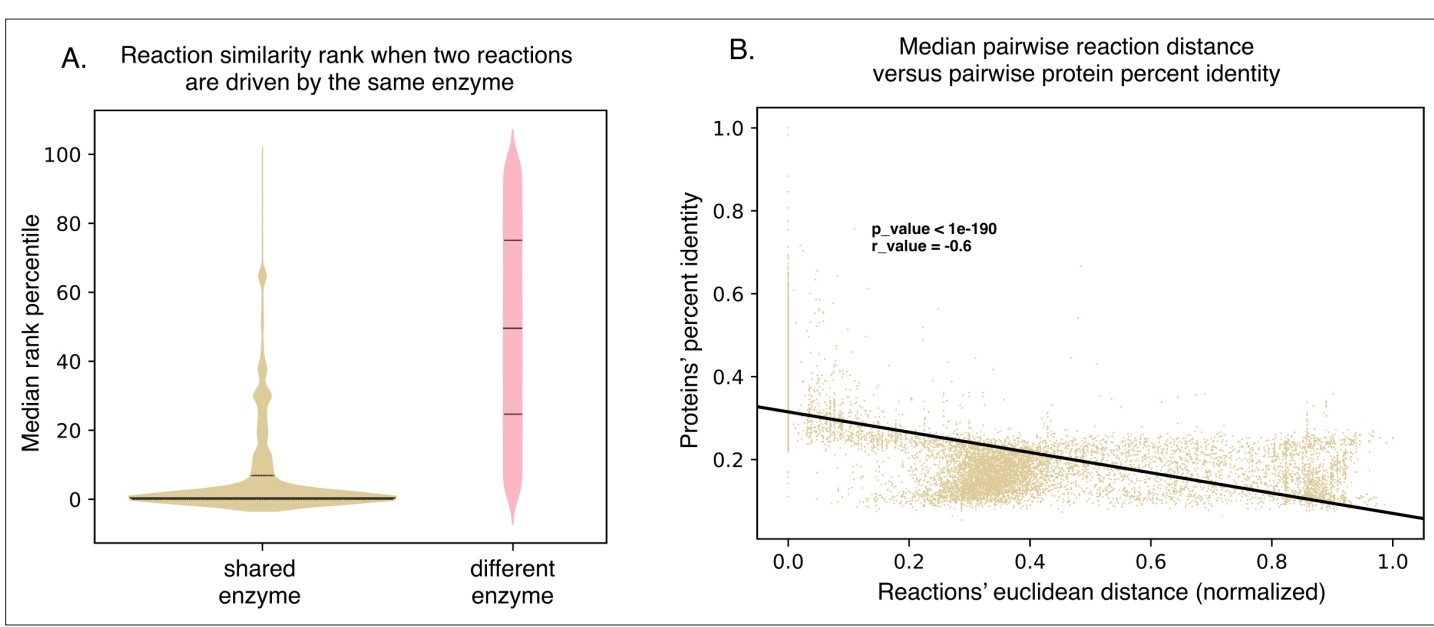

**Figure 3.** SIMMER's chemical representations can be used to find shared, responsible enzymes. (**A**) When SIMMER was queried with a MetaCyc reaction, other reactions driven by the same enzyme are returned as the most similar. As a contrast, reactions driven by a different enzyme yield a more uniform rank distribution. Solid lines of the violin plots depict median reaction similarity rank and dashed lines represent lower and upper quartile ranges. (**B**) There is a negative association between pairwise reaction euclidean distance and pairwise protein global percent identity, but it is highly variable, demonstrating the need for protein searches that do not rely on high global percent identity to capture shared function.

The online version of this article includes the following source data for figure 3:

**Source data 1.** Pairwise similarities for SIMMER's underlying chemical and protein data.

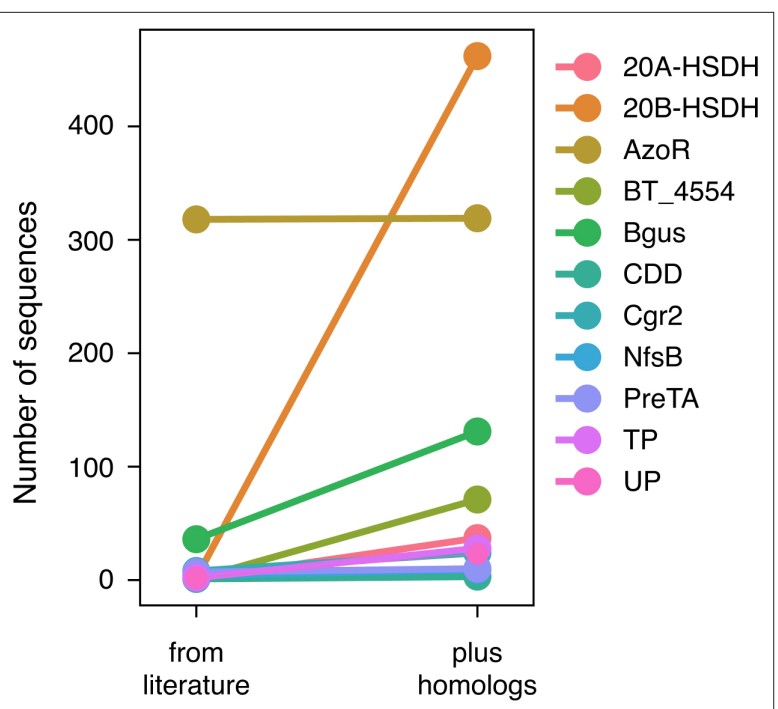

**Figure 4.** An expanded list of gut bacterial enzymes relevant to known cases of drug metabolism. Eleven of the 18 enzymes responsible for positive control drug metabolism events have high-confidence homologs that we gathered by filtering for biological significance.

The online version of this article includes the following source data and figure supplement(s) for figure 4:

**Source data 1.** UHGG database identifiers for all putative homologs of the 18 previously characterized drug metabolizing enzymes.

**Figure supplement 1.** An expanded list of gut bacterial enzymes relevant to known cases of drug metabolism.

without a shared enzyme (*Figure 3A*). We also found a negative association between chemical reaction distance and global sequence similarity of MetaCyc enzyme annotations, though there is much variation in this relationship (*Figure 3B*). This reflects the known association between sequence similarity and similarity in chemical function, as well as reports that this relationship can often be overestimated (*Tian and Skolnick, 2003*). Indeed, MetaCyc reactions with identical chemical representations could be conducted by proteins with only ~20% global percent identity all the way to 100% (*Figure 3B*). This relationship informed our decision to use pHMM searches rather than BLAST to find sequence similar proteins from the human gut microbiome, as pHMM searches rely on matches of conserved sites rather than global percent identity between sequences. Together, these analyses informed the manner in which we combined chemical and protein similarity to create a microbiome enzyme prediction tool.

## An expanded list of gut bacterial enzymes relevant to known cases of drug metabolism

To assess SIMMER's prediction accuracy for previously characterized reactions and to mount a comparison to existing methods, we used drug metabolism as a use-case. First, we curated 300 drug metabolism events associated with the human gut microbiome from the literature (*Supplementary file 1*). For 33 of these reactions the responsible bacterial enzyme, characterized metabolite(s), and associated EC annotation are known (*Supplementary file 1*). These 33 reactions are conducted by 18 enzymes. Due to orthology and proclivity for genetic transfer between even distantly related bacteria, however, there are likely many as yet undiscovered homologs of these drug-metabolizing enzymes that can catalyze identical drug metabolism events (*Pollet et al., 2017*). To account for this, we created an expanded database (*Figure 4*, *Figure 4—figure supplement 1*, *Figure 4—source data 1*) of the 18 characterized enzymes from pHMM and phmmer searches of the UHGG database (*Almeida et al.,*

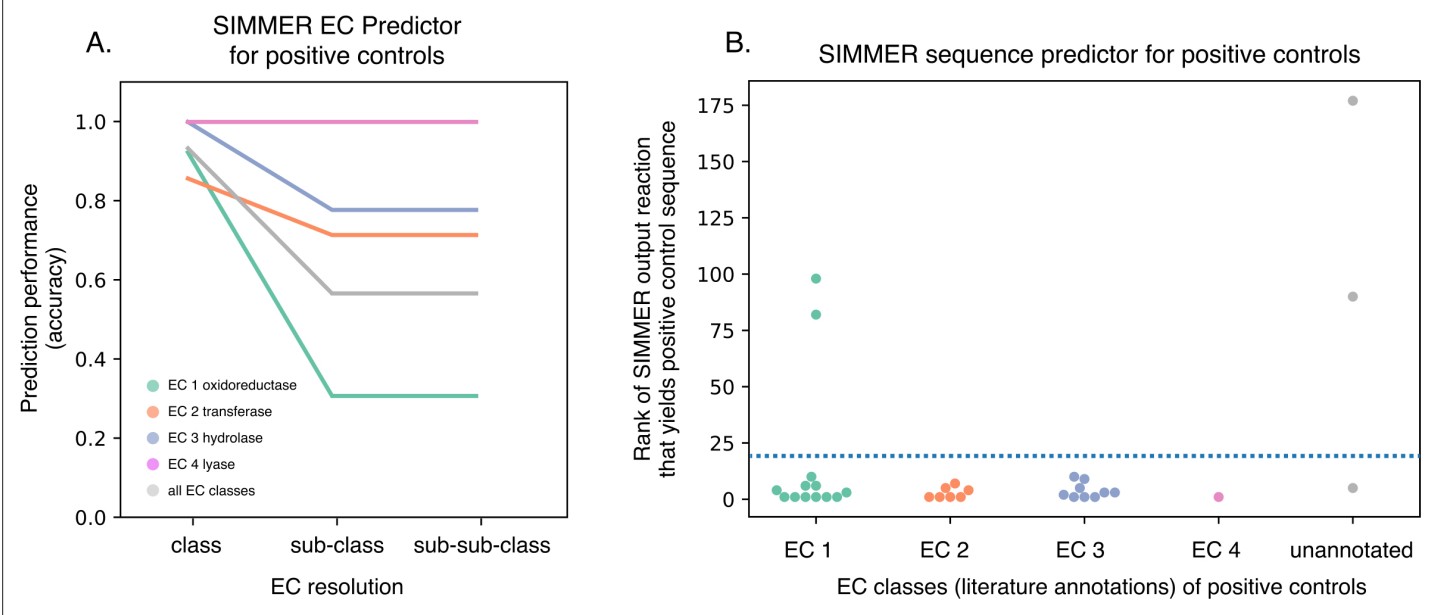

**Figure 5.** SIMMER captures known gut bacterial enzymes involved in drug metabolism. (**A**) SIMMER accurately predicted Enzyme Commission (EC) classes for 28 previously characterized reactions that possess EC annotations. As with the MetaCyc database (*Figure 2B*), accuracy dropped off as EC resolution increased. (**B**) SIMMER predicted bacterial sequences previously shown to drive 33 drug metabolism events in the gut microbiome. Depicted is the rank (out of $N = 8914$ reactions) of the MetaCyc reaction that yielded a gut microbiome homolog matching the known positive control sequence. Reported accuracy is based on such a reaction being within the top 20 ranked reactions (dashed blue line).

The online version of this article includes the following source data and figure supplement(s) for figure 5:

**Source data 1.** Performance analysis of 33 known chemical transformations (positive controls) using SIMMER's full representation or only a substrate.

**Figure supplement 1.** Distributions of all MetaCyc reactions' euclidean distances to the positive control list queries.

**Figure supplement 2.** The relationship between accuracy and false positives as a user explores more reactions from SIMMER's prediction output.

---

*2021*), yielding 52,849 total candidate homologs (a median of 1087 candidates per enzyme). After filtering enzymes by hmmer significance, alignment length, presence in data from the human jejunum (*Zmora et al., 2018*) and RNA-sequencing studies (*Integrative HMP (iHMP) Research Network Consortium, 2019*), and predicted affinity for the substrate in question using the Similarity Ensemble Approach (*Keiser et al., 2007*), our database contained a median of 2 high-confidence homologous sequences per enzyme (range = 0–460 across the 18 enzyme families, *Figure 4*, *Figure 4—figure supplement 1*, *Figure 4—source data 1*). These 741 additional enzyme sequences for 33 reactions formed an expanded testing set of known gut bacterial enzymes.

## SIMMER captures known gut bacterial enzymes involved in drug metabolism

With our expanded database of drug-metabolizing enzymes from the human gut microbiome in hand, we next verified that SIMMER can accurately predict responsible enzymes and reaction types for the 33 known chemical transformations. Only 3 of these 33 reactions are themselves MetaCyc entries (5-ASA, dopamine, and levodopa degradation); if enzymes of reactions not described in MetaCyc were also accurately predicted, it would show that SIMMER can discover non-identical yet chemically similar reactions.

SIMMER accurately predicted the specific enzymes from the human gut microbiome that conduct the 33 query reactions (*Figure 5—source data 1*). This enzyme list was populated by the results of the precomputed pHMM searches of human microbiome catalogs with annotated gene sequences from MetaCyc reactions (*Figure 1A*). In 29 cases (88%), the characterized (i.e., positive control) enzyme(s) for a reaction was found in the output enzyme list for the top 20 of the ranked MetaCyc reactions (*Figure 5*, *Figure 5—figure supplement 1*). We chose the top 20 out of ~9000 reactions as an accuracy cutoff, because this was the median rank of a true positive reaction in our analysis of MetaCyc

reactions conducted by the same enzyme (*Figure 3A*, *Figure 5—figure supplement 2*). Since the positive controls span four EC classes (EC1 oxidoreductases, EC2 transferases, EC3 hydrolases, and EC4 lyases), this result demonstrates SIMMER's ability to accurately predict microbiome-based enzymes for a diversity of reaction types. Also, despite inaccurate EC predictions for nicardipine reduction and brivudine transformation, SIMMER was able to, respectively, predict AzoR and BT_4554 enzymes as responsible for the reactions, because enzyme and EC predictions are separately computed by SIMMER (*Figure 5*, *Figure 5—figure supplement 1*, *Figure 5—source data 1*).

SIMMER often predicts hundreds of sequences potentially responsible for an input reaction, meaning that the possibility of false positives is high. Because biotransformations of the microbiome are understudied, however, there does not exist in the literature a definitive list of *all* the bacterial enzymes that do *not* conduct a given reaction. For this reason, we were unable to directly assess an enzyme prediction false positive rate. Instead, we took a conservative approach and defined all output reactions preceding that which yielded the positive control enzyme sequence as false positives (*Figure 5—figure supplement 2*).

Of the 33 drug metabolism events known to occur via human gut bacterial enzymes, EC annotations exist for 30. For five queries SIMMER predicted more than one significant EC class, but for 28 out of the 30 reactions (including these five), SIMMER's top EC class prediction was correct (*Figure 5*, *Figure 5—figure supplement 1*, *Figure 5—source data 1*). The two failed EC predictions were for nicardipine reduction (inappropriately predicted as an isomerase reaction) and for brivudine transformation (for which SIMMER made no significant prediction).

**Table 1.** Comparison to existing methods.

| | | DrugBug (Sharma et al., 2017) | MicrobeFDT (Guthrie et al., 2019) | SIMMER |
|---|---|---|---|---|
| Underlying databases | Number of chemicals | 2324 compounds | 10,822 compounds | 8914 reactions (12,439 unique compounds) |
| | Number of bacterial genomes | 491 (custom database) | 3008 (IMG) | 286,997 (UHGG) |
| Input types | Accepts novel SMILES | | | ✓ |
| | User options for different chemical fingerprints | ✓ | | ✓ |
| Output types | Reaction similarity measure | | | ✓ |
| | EC predictions | ✓ | ✓ | ✓ |
| | Enzyme and species predictions | ✓ | | ✓ |
| | Function predictions | | | ✓ |
| | Prevalence/abundance | | ✓ | ✓ |
| | Network relationships | | ✓ | |
| Usability | Web server | ✓ | | ✓ |
| | Command-line tool | | | ✓ |
| | Docker container | | ✓ | |
| Accuracy | Prediction of previously characterized enzymes | 3% | NA | 88% |
| | Prediction of previously characterized EC numbers | 37% | 29% | 93% |

The online version of this article includes the following source data for table 1:

**Source data 1.** All data relevant to the comparison of SIMMER, DrugBug, and MicrobeFDT methods.

## SIMMER outperforms existing methods

To mount a comparison to the other in silico methods that, in part, aimed to describe microbiome drug metabolism, we next queried the 33 positive control reactions using MicrobeFDT and DrugBug (*Table 1*) both of which rely solely on substrate chemical similarity rather than information from a whole reaction (*Guthrie et al., 2019*; *Sharma et al., 2017*).

For the 30 EC annotated positive control reactions, DrugBug had 37% accuracy in predicting EC classes (in comparison to SIMMER's 93%) and predicted the correct enzyme for only a single reaction, SN38 glucuronide deconjugation, despite the presence of chemically similar reactions metabolized by the same enzyme amongst the positive controls (*Table 1—source data 1*). We additionally queried SIMMER with the four positive controls (ginsenoside Rb1, quercetin-3-glucoside, cycasin, and sorivudine) associated with characterized bacterial enzymes from the original DrugBug publication (*Table 1—source data 1*). Both DrugBug and SIMMER were able to predict EC classes for sorivudine, but only SIMMER was able to accurately predict the specific enzyme (BT_4554) responsible for the drug's degradation. For ginsenoside Rb1 (3.2.1.192), quercetin-3-glucoside (3.2.1.21), and cycasin (3.2.1.21), SIMMER accurately predicted EC codes out to sub-sub-class (3.2.1.-), serial designation (3.2.1.21), and sub-sub-class (3.2.1.-), respectively, which was a resolution improvement over DrugBug (*Table 1—source data 1*).

We next queried the 30 EC annotated drug metabolism positive controls against MicrobeFDT, a chemical graph tool that only predicts EC codes, not enzyme sequences. MicrobeFDT produced EC predictions for 14 of the 30 positive controls, four of which were correct (29% accuracy). We finished the comparison between SIMMER and MicrobeFDT by querying SIMMER with the metabolism use-case described in the MicrobeFDT publication, altretamine demethylation. In our hands, there was no Cypher query against the MicrobeFDT database that resulted in a demethylase EC code (we determined possible demethylase EC codes by running a query in the Swiss Institute for Bioinformatics Enzyme Nomenclature Database) (*Bairoch, 2000*). We performed queries of direct EC annotation for melamine and altretamine, as well as EC annotation queries for any compound with either substructure or toxicity overlap with altretamine or melamine. The closest result to a demethylase enzyme was a cypher query of toxicity overlap with altretamine that yielded a nitric oxide synthase (EC 1.14.13.39) acting on L-arginine among its results (*Table 1—source data 1*). For its significant EC (reaction type) prediction, SIMMER identified altretamine demethylation appropriately as an oxidoreductase reaction acting on a CH-NH group of donors (EC 1.5.-), but not significantly as a demethylation event (*Table 1—source data 1*).

Neither DrugBug nor MicrobeFDT computed predictions for all 33 positive control reactions. MicrobeFDT made so few EC predictions (14/30) because it is reliant on a fixed chemical database that cannot be modified by the user, meaning that a compound cannot be queried if it is not already present in MicrobeFDT's graph. While DrugBug does allow for novel chemical input, its protein database only contains information from 491 bacterial genomes (in comparison to SIMMER's 286,997 genomes), affecting its ability to make enzyme-specific predictions.

In instances when DrugBug and MicrobeFDT did make predictions, they suffered from low accuracy (*Table 1*), which we hypothesized was due to both methods' reliance on substrate rather than reaction chemistry. Biotransformations involve the relationship between substrate(s), cofactor(s), and an enzyme to yield a particular product(s). As one substrate can exhibit affinity for multiple enzymes, resulting in multiple unique products, sole employment of substrates in a chemical fingerprint does not achieve the resolution necessary to make relevant predictions. To test if SIMMER's better performance could be attributed to including cofactors and products, we modified our code to run with a chemical representation that includes only the substrate of each positive control reaction. Enzyme prediction accuracy dropped from 88% down to 33%, and EC prediction accuracy dropped from 93% down to 48% (*Table 1—source data 1*), supporting the hypothesis that SIMMER's better performance when compared to DrugBug and MicrobeFDT is due in large part to our use of chemical representations that include the full reaction. These results are in line with our previous demonstration that SIMMER clusters enzymatic reaction chemistry only when a full reaction is employed (*Figure 2*, *Figure 2—figure supplement 4*).

Altogether, these findings illustrate SIMMER's enhanced accuracy over other methods for the use-case of characterized drug metabolism events by gut bacteria, and also illustrate SIMMER's novel ability to predict species and enzymes responsible for chemical transformations not previously described in literature or databases.

## SIMMER predicts novel drug-metabolizing enzymes

After establishing SIMMER's accuracy in predicting drug-metabolizing enzymes in the human gut environment, we predicted EC codes, functional annotations, and enzyme sequences for novel microbiome drug metabolism reactions that do not yet possess a responsible, characterized enzyme (*Figure 6—source data 1*). From our literature curation of 300 non-antibiotic therapeutics affected by the microbiome (*Supplementary file 1*), we were confident that 88 are directly metabolized by gut bacteria due to their association with an identified bacterial metabolite in the literature. We formatted these 88 reactions in SMILES format and input them as queries to SIMMER.

Of the 88 reactions queried, SIMMER determined significant EC predictions for 75 reactions (86.2%), and 61 (70.1%) of these were out to the serial designation (i.e., highest resolution) EC code (*Figure 6—source data 1*). This list of 61 transformations presents reactions for which we believe enzyme characterization is worth pursuing as our predictions are significantly similar to enzymes already explored in the literature. SIMMER's EC predictions resulted in expanded and diversified EC class membership for drug transformations known to occur in the microbiome (*Figure 6C*). Of interest, this analysis resulted in a large expansion of putative hydrolysis, reduction, and isomerization reactions in the human gut microbiome. The number of SIMMER predictions varies widely by reaction, with median output of 372 genes, 286 genomes, and 10 phyla predicted as responsible across the 88 reactions (*Figure 6—source data 1*, *Figure 6A*). Unsurprisingly, many of these reactions are predicted to occur due to enzymes found in Firmicutes, Bacteroidetes, Actinobacteria, and Proteobacteria, but there are also SIMMER enzyme predictions in phyla not previously associated with drug metabolism (*Figure 6B*).

## Experimental validation of SIMMER predictions: MTX case study

Among the 88 uncharacterized reactions, we assessed SIMMER's ability to predict strains and enzymes for the hydrolysis of MTX. Oral MTX is an anti-folate immunosuppressant, and the first-line therapy for individuals with rheumatoid arthritis (RA). About half of RA patients, however, show inadequate response to MTX (*Scher et al., 2020*), and the reasons for this remain incompletely understood. This lack of efficacy may be due to altered pharmacokinetics, as oral bioavailability of MTX is subject to large inter-individual variation with values ranging from 32% to 70% (*Roon and Laar, 2006*). MTX is estimated to undergo extensive enterohepatic circulation, which results in increased exposure to gut bacteria in the small intestine (*Roon and Laar, 2006*). This exposure may result in MTX depletion, as it is known that MTX degrades to inactive metabolites 2,4-diamino-$N^{10}$-methylpteroic acid (DAMPA) and glutamate in mice with intact gut microbiomes, but not in their antibiotic treated counterparts (*Zaharko et al., 1969*). In a recent human study, ex vivo stool samples from MTX non-responders depleted MTX to a higher degree, though the relevant strains and enzymes were not determined (*Artacho et al., 2021*).

When queried with MTX and its gut bacteria associated metabolites DAMPA and glutamate, SIMMER calculated a most similar MetaCyc reaction (3.4.17.11-RXN) and a significant EC prediction (3.4.17.11, p-value <0.001). This MetaCyc reaction describes the hydrolysis of folate into pteroate and glutamate, driven by a glutamate carboxypeptidase (Cpg2) found in environmental *Pseudomonas aeruginosa*. Hydrolysis of MTX is chemically similar to hydrolysis of folate (*Figure 7A*) with a Tanimoto coefficient = 0.6, and normalized euclidean distance = 0.05 in SIMMER's precomputed chemical space. SIMMER made 2286 human gut microbiome enzyme predictions for degrading MTX into DAMPA and glutamate (*Figure 7—source data 1*). The most common Prokka annotation for these enzymes is Carboxypeptidase G2s (*Figure 7B*) due to their sequence similarity to MetaCyc's environmental Cpg2. Furthermore, SIMMER's predicted enzymes had a median global identity of 33% to *Pseudomonas* sp. RS-16 (an environmental bacterium) Cpg2, an enzyme known to conduct hydrolysis of MTX (*Larimer et al., 2014*; *Jeyaharan et al., 2016*). This similarity suggests that some of SIMMER's predicted enzymes conduct hydrolysis of MTX, similar to *P. aeruginosa's* Cpg2 reaction.

We next screened an existing collection of 42 diverse bacterial strains found in the human gut for its ability to deplete MTX. This collection was of interest due to previously determined inter-strain variation in growth inhibition by MTX (*Nayak et al., 2021*). Each isolate was incubated with MTX (100 µg/ml), and MTX levels were determined by high-performance liquid chromatography (HPLC) (*Figure 8*). MTX depletion varied across the strain collection, with 10 isolates exhibiting at least a 50% decrease in MTX levels compared to control. There was a statistically significant concordance

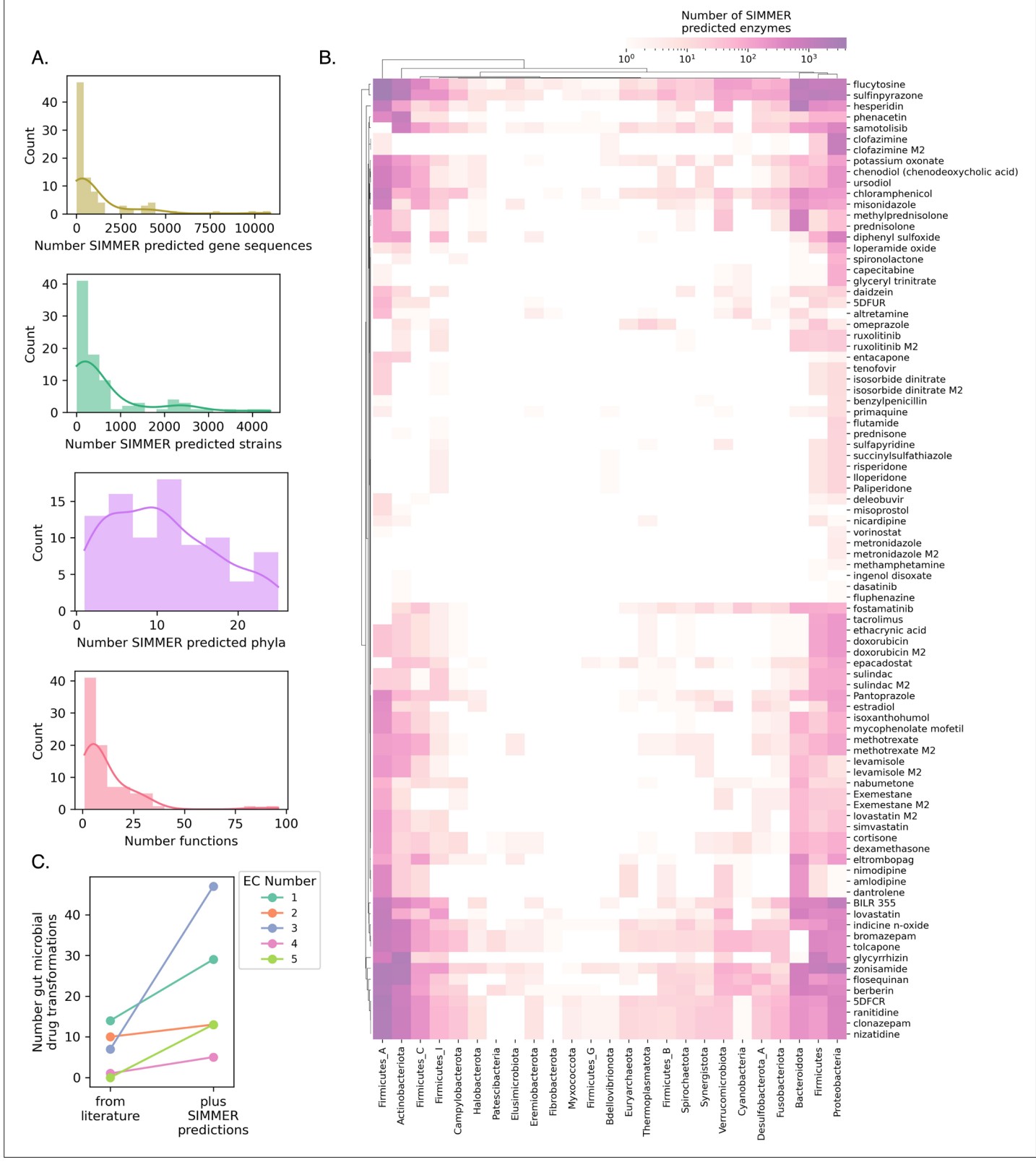

**Figure 6.** SIMMER predicts novel drug metabolizing enzymes. (**A**) Distributions depict the unique number of genes, strains, and phyla predicted to be responsible for 88 reported drug transformation reactions, as well as predicted gene functions. (**B**) A heatmap illustrating, for a given phylum, the number of unique drug-metabolizing enzymes predicted to conduct 88 different drug metabolism events. (**C**) Enzyme Commission Class representation

*Figure 6 continued on next page*

*Figure 6 continued*

for bacterial transformations of therapeutics before and after the employment of SIMMER. Our predictions greatly expand the number of characterized reduction (EC1), hydrolysis (EC2), and isomerization (EC5) events and modestly increase the number of transferase (EC2) and lyase (EC4) events.

The online version of this article includes the following source data for figure 6:

**Source data 1.** SIMMER's enzyme predictions (with phyogenetic annotations) and EC predictions for 88 previously uncharacterized reactions.

---

between SIMMER's predictions of which strains had an enzyme capable of metabolizing MTX and our experimental results (Fisher's exact test, odds ratio = 5.4, p-value <0.05, *Figure 8*). We next examined the seven strains for which SIMMER predicted capability to perform MTX hydrolysis but where we did not observe decreased drug levels. One of these discordant strains, *E. coli* BW25113, possesses the multidrug resistance efflux pump AcrAB-TolC, which actively exports MTX (*Kopytek et al., 2000*), explaining why MTX was not affected. To test if other discordant strains possessed orthologs of this efflux pump, we looked for AcrAB sequences in the genomes of all 42 bacterial strains. We found two more discordant strains with AcrAB genes: *Edwardsiella tarda* and *Providencia rettgeri* (*Figure 8*). Thus, 3/7 SIMMER predictions that did not match experimental results could be explained by the ability of drug efflux to mask the potential for enzymatic activity.

As an additional comparison to DrugBug and MicrobeFDT, we assessed MTX metabolism with both tools. Both incorrectly predicted that MTX degradation is through transferases rather than hydrolases. Because DrugBug also provides species and enzyme predictions, we assessed its ability to predict species responsible for MTX metabolism. We found that only SIMMER was able to significantly predict species capable of MTX metabolism (SIMMER odds ratio = 5.4 versus DrugBug odds ratio = 0). DrugBug produced zero true positives and predicted that MTX metabolism was a feature largely restricted to members of Bacteroidetes, a group who exhibited no activity in our in vitro assay (*Figure 8—figure supplement 1*).

## Evaluation of SIMMER on external datasets

We next sought to determine the relationship between abundance of SIMMER predicted MTX-metabolizing enzymes and clinical response in patients. We took advantage of a publicly available dataset in which fecal samples from treatment-naive new-onset RA patients were profiled using shotgun sequencing; these same patients were followed for 4 months and MTX responder status and disease activity (DAS28) were assessed by the authors (*Artacho et al., 2021*). Response was defined as a DAS28 score improvement of at least 1.8 points at 4 months of MTX therapy (*Artacho et al., 2021*). Of SIMMER's 2286 enzyme predictions for MTX hydrolysis, 386 were detected in the fecal samples of RA patients at baseline (prior to treatment). MTX non-responders exhibited a significant enrichment of SIMMER MTX predictions in their stool samples (*Figure 9A*). Similarly, a significant negative correlation was seen between patients' disease score improvements and abundance of SIMMER enzyme predictions (*Figure 9B*). These data, combined with our HPLC validation and prior ex vivo incubations linking non-response to increased MTX metabolism (*Artacho et al., 2021*), emphasize the utility of using SIMMER to uncover clinically relevant drug–microbe interactions.

Next, we sought experimental evidence for other novel transformations with SIMMER predictions. The side-chain cleavage of dexamethasone to 17-oxodexamethasone is one example. Dexamethasone was recently shown to be metabolized solely by *Clostridium scindens* (ATCC 35704) out of a collection of 76 isolates representative of the human gut microbiome (*Zimmermann et al., 2019b*). When the authors assessed dexamethasone metabolism in 28 shotgun sequenced human stool samples, metabolite formation varied substantially by individual, but could not be explained by *C. scindens* species abundance. To explore this lack of correlation in light of our findings, we assessed the abundance of *C. scindens* SIMMER enzyme predictions (*Figure 10—source data 1*) within each of the 28 samples (i.e., the number of *C. scindens* SIMMER enzyme predictions aligned to each sample's shotgun sequencing reads) rather than each sample's *C. scindens* species relative abundance. We found a significant association between metabolite formation and number of SIMMER enzyme aligned reads, and also a significant association between parent compound consumption and number of SIMMER enzyme aligned reads (*Figure 10B*). This underscores the importance of identifying an enzyme and quantifying its presence in metagenomic data.

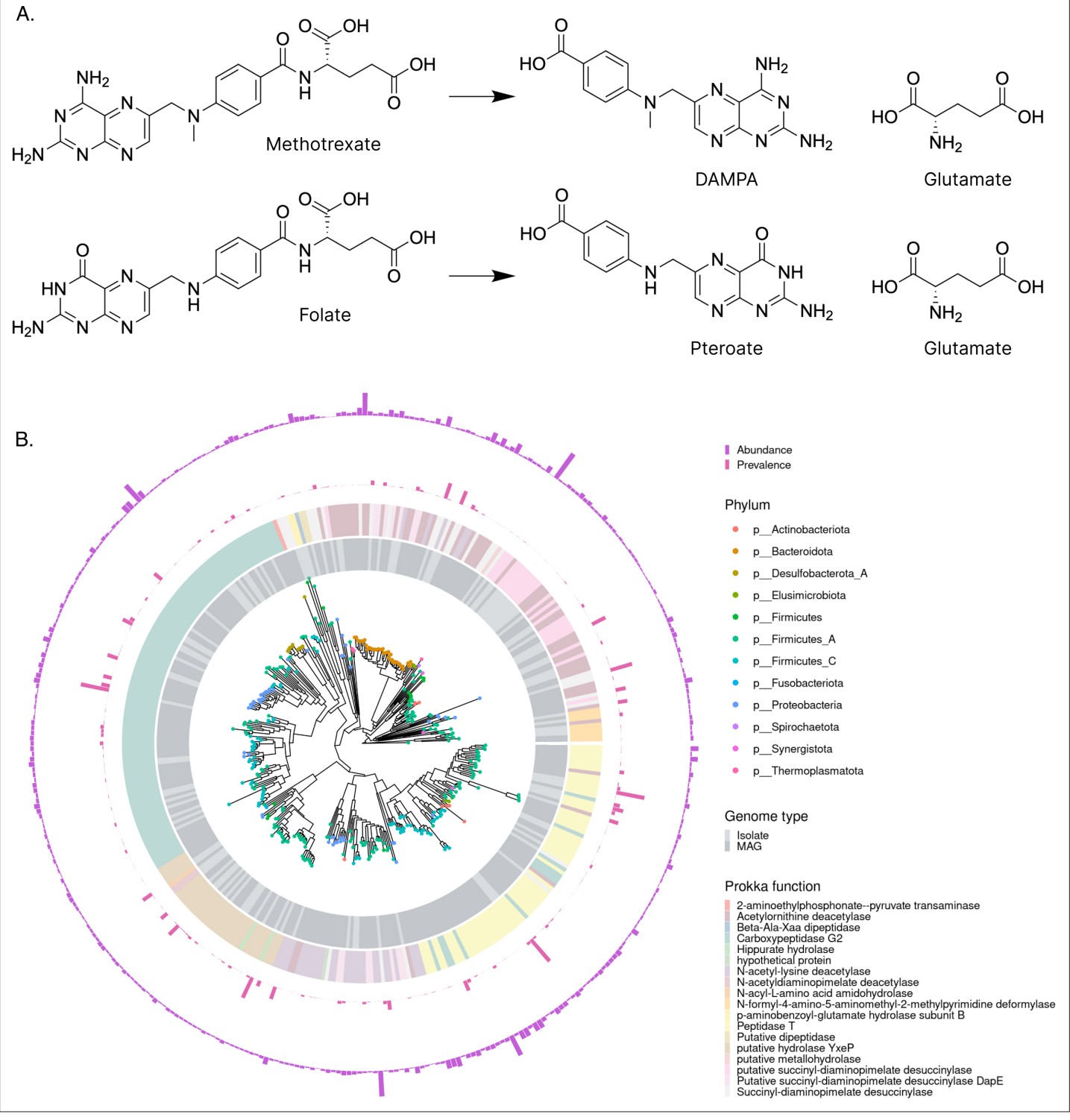

**Figure 7.** SIMMER predicts methotrexate (MTX) metabolizing enzymes similar to known environmental MTX metabolizers. (**A**) When queried with MTX hydrolysis to 2,4-diamino-$N^{10}$-methylpteroic acid (DAMPA) and glutamate, SIMMER found that folate hydrolysis was the most chemically similar MetaCyc reaction. (**B**) SIMMER predicted 2286 unique bacterial sequences putatively capable of MTX hydrolysis to DAMPA and glutamate. There was great variability in the prevalence and abundance of these sequences in healthy human metagenomic data. Among the strains with predictions, Firmicutes were most common. The most frequent Prokka annotation was Carboxypeptidase G2.

The online version of this article includes the following source data for figure 7:

**Source data 1.** SIMMER's enzyme prediction output table for MTX hydrolysis to DAMPA and glutamate: 2,286 bacterial sequences and their UHGG database identifiers, taxonomy, predicted function, and prevalence and abundance in healthy humans.

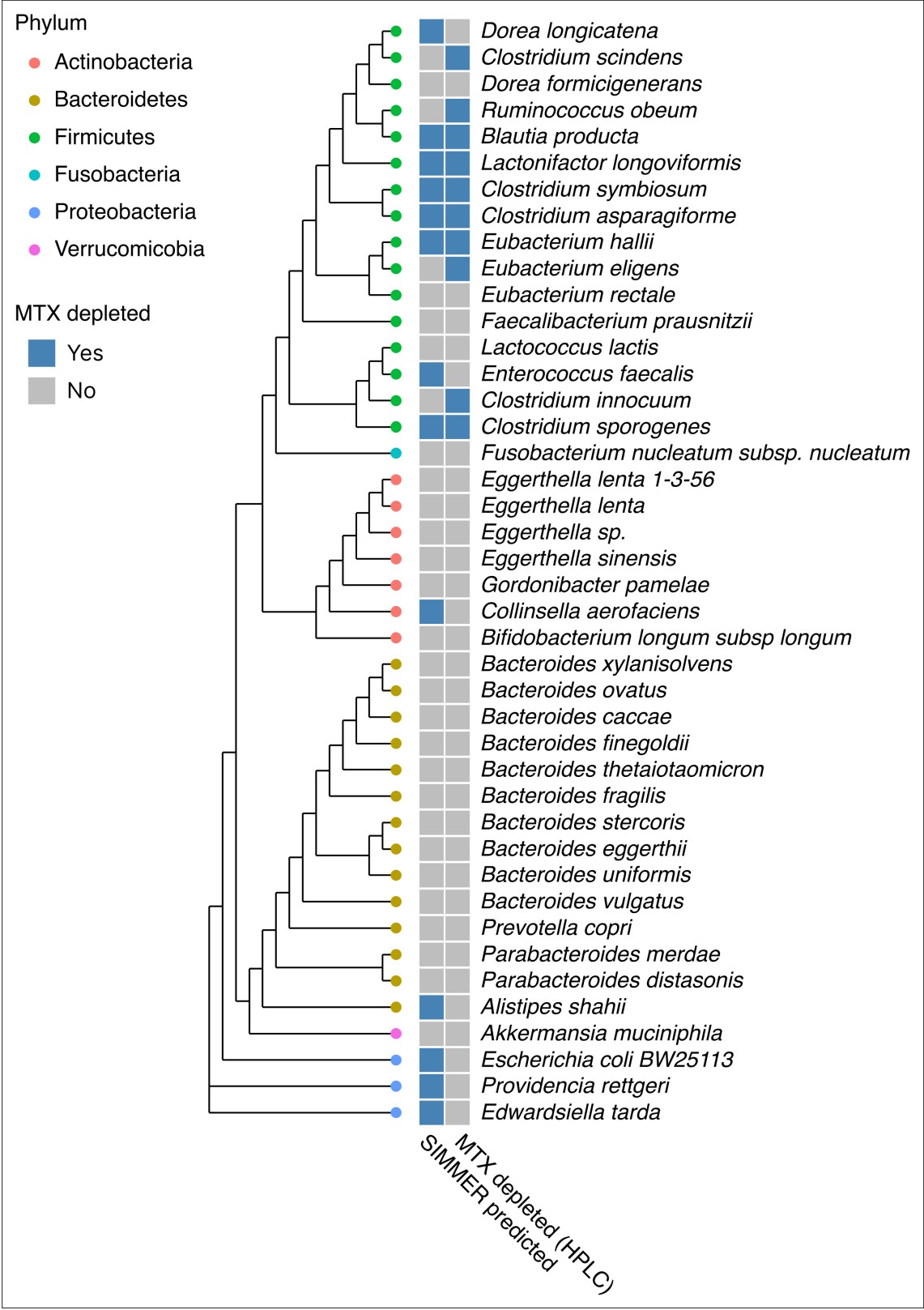

**Figure 8.** SIMMER accurately predicted bacterial strains capable of methotrexate (MTX) depletion. A diverse panel of 42 isolates was incubated with MTX, and depletion (yes/no) measured via HPLC (50% decrease relative to sterile controls). SIMMER predicted (yes/no) that 13 of the 42 isolates were capable of MTX metabolism, and HPLC experiments showed that 10 isolates depleted MTX (SIMMER prediction p-value = 0.046, Fisher's exact test).

*Figure 8 continued on next page*

*Figure 8 continued*

The maximum likelihood phylogenetic tree (PhyML) was created using 16S rRNA gene sequences from the 42 organisms' closest representatives in the Greengenes database.

The online version of this article includes the following source data and figure supplement(s) for figure 8:

**Source data 1.** Identifiers and assembly file names for 42 isolates incubated with MTX, and example HPLC traces from two MTX depleting strains.

**Figure supplement 1.** DrugBug methotrexate (MTX) metabolism predictions.

**Figure supplement 2.** Representative methotrexate (MTX) depletion high-performance liquid chromatography (HPLC) traces.

It came to our attention while preparing this manuscript that recombinant steroid-17,20-desmolase (DesAB) enzymes from *C. scindens* were shown to perform side-chain cleavage on prednisone, but also to a lesser extent on dexamethasone. DesAB's reduced activity for dexamethasone was assumed to be due to the compound's potentially inhibitory 16α-methyl group (*Ly et al., 2020*). To ensure that SIMMER's enzyme prediction for dexamethasone cleavage was not enriched in metabolizing stool samples due to co-occurrence with already known DesAB, we next assessed the abundance of *desAB* reads across the 28 samples and found no significant correlation between number of reads and either metabolite formation or dexamethasone consumption slopes (*Figure 10—figure supplement 1*). These results indicate that species-level information alone is not enough to predict chemical transformations in a microbiome sample, but with SIMMER, knowledge of responsible enzymes can recapitulate a sample's potential for therapeutic degradation.

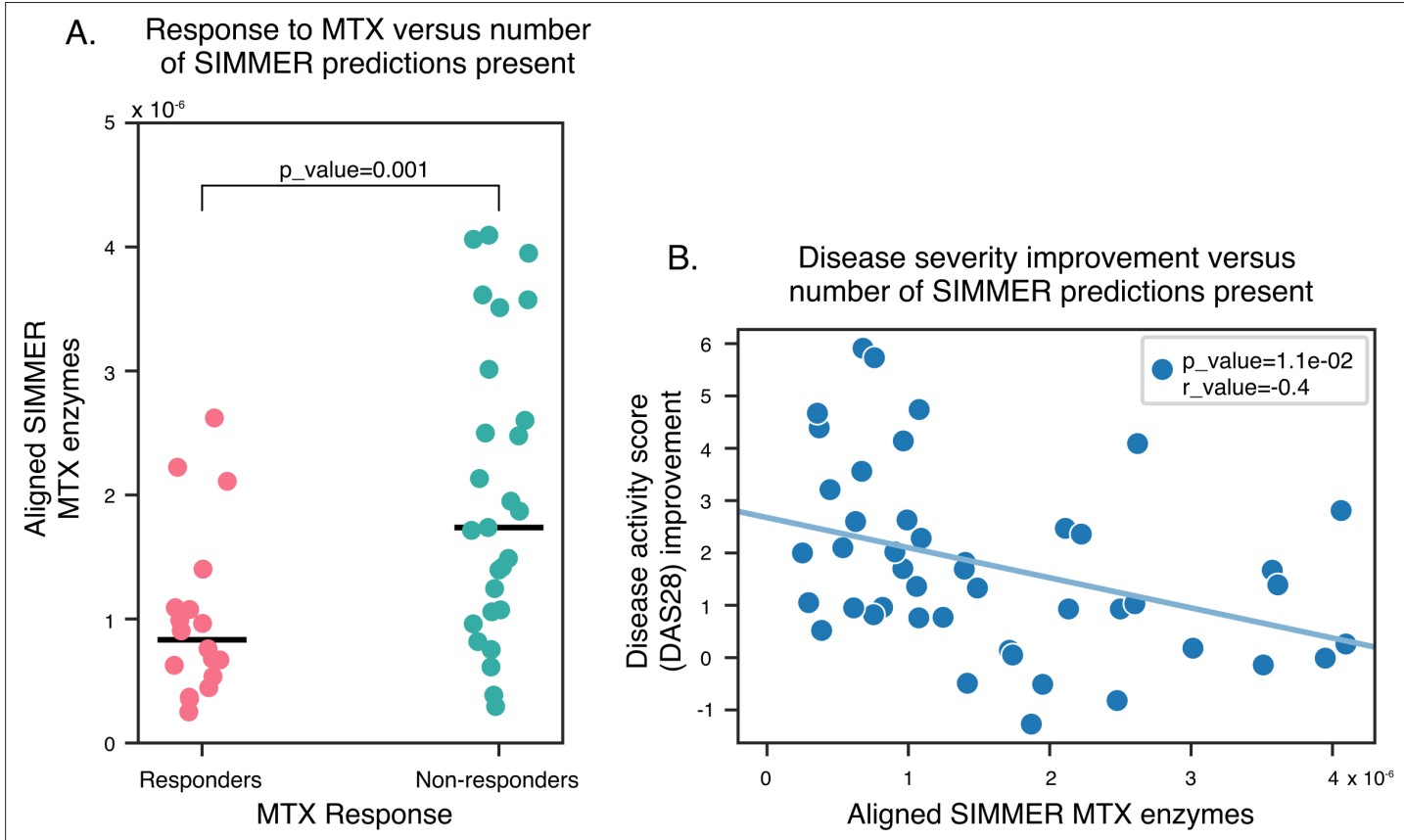

**Figure 9.** SIMMER enzyme predictions are enriched in methotrexate (MTX) non-responders. *N* = 368 of SIMMER predicted sequences for MTX hydrolysis were found in new-onset RA patients with variable MTX response. (**A**) SIMMER enzyme predictions (normalized by read depth) were enriched in MTX non-responders (Generalized Linear Model (GLM) Poisson, p-value = 0.001), and (**B**) a significant negative association between disease severity improvement and number of SIMMER enzyme predictions (normalized by read depth) was observed (Pearson correlation = −0.4, Student's *t*-test p-value = 0.01).

The online version of this article includes the following source data for figure 9:

**Source data 1.** Disease activity (DAS28) scores, MTX response status, and number of SIMMER predicted enzymes for RA patients.

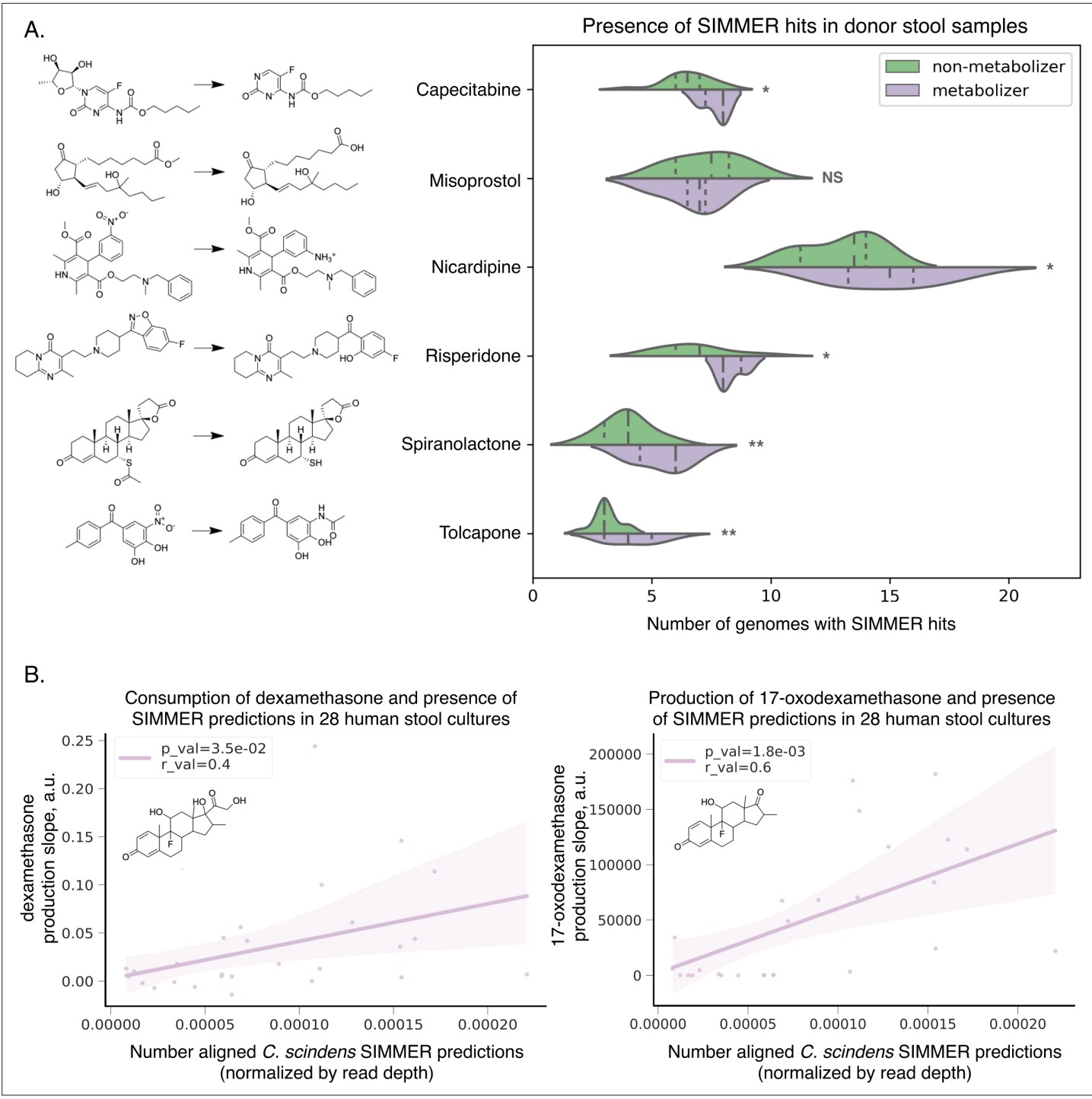

**Figure 10.** SIMMER predicted enzymes explain inter-individual variations in drug metabolism. (**A**) Donors (*N* = 20) from the Javdan et al. 16S rRNA gene sequencing study (***Javdan et al., 2020***) possessed an enrichment of genomes harboring SIMMER enzyme predictions when metabolism of a given drug was observed. Violin plot curves were made using a seaborn package that performs a kernel density estimation of the underlying datapoint distribution. Chemical transformations were drawn using ChemDraw software. Single asterisks denote p-values ≤0.05, and double asterisks denote p-values ≤0.01 (*t*-test). (**B**) There was a significant correlation between a human stool sample's ability to consume dexamethasone (consumption slope, a.u.), to produce 17-oxodexamethasone (production slope, a.u.), and the number of aligned SIMMER predicted sequences for side-chain cleavage of dexamethasone. Patient (*N* = 28) conversion slopes and metagenomics data were accessed from the original study (***Zimmermann et al., 2019b***). Chemical structures were drawn using ChemDraw software.

The online version of this article includes the following source data and figure supplement(s) for figure 10:

*Figure 10 continued on next page*

*Figure 10 continued*

**Source data 1.** SIMMER predictions for previously published drug metabolism studies.

**Figure supplement 1.** Dexamethasone metabolism is not significantly correlated with presence of *desAB*.

Lastly, eight of the 88 novel transformations were among those investigated in a high-throughput study exploring the metabolism of 571 compounds in ex vivo stool samples (*Javdan et al., 2020*). This publication demonstrated bacterial degradation of 57 therapeutics in a single pilot donor stool sample (with associated shotgun sequencing), as well as in 20 human stool samples (with associated 16S rRNA gene sequencing). While this study greatly expanded the number of drugs known to break down in the presence of gut bacteria and identified eight metabolite structures, it only identified a responsible enzyme in two of the 57 drug degradation cases due to the low-throughput nature of enzyme characterization. To further assess SIMMER's ability to predict novel enzymes, and to demonstrate the utility of using SIMMER in an experimental context, we investigated the presence of our predictions in the Javdan et al. study sequencing results. Because shotgun metagenomics sequencing for the pilot donor was deposited, we were able to confirm via tBLASTn searches that SIMMER enzyme predictions were directly found in the pilot donor stool sample for all eight of the reactions with identified metabolites (*Figure 10—source data 1*). However, the sequencing data from the 20 human donor study were only 16S profiling, so we were unable to look directly for SIMMER enzyme predictions. We instead ensured that genomes found in metabolizing stool samples were SIMMER species predictions. We found that donors who could metabolize a given drug possessed a significant enrichment of species predicted by SIMMER. This was the case for five out of the six reactions analyzed (*Figure 10—source data 1*, *Figure 10A*). This result validates SIMMER's ability to predict species potentially able to perform a queried metabolic reaction, though additional data would be needed to confirm that the strains in these 20 donors had the predicted enzyme.

## SIMMER software

In addition to providing SIMMER (https://github.com/aebustion/SIMMER, copy archived at *Bustion, 2023*) as a command-line tool that quickly generates enzyme sequence predictions (fasta and tab-separated-value files), EC predictions (tab-separated-value file), and MetaCyc reactions ranked by similarity (tab-separated-value file) based on a user's input reaction, SIMMER is also available as a user-friendly website (https://simmer.pollard.gladstone.org/). The user can either input one query reaction at a time or upload multiple reactions in tsv file format (*Figure 11*). All output types available with the SIMMER command-line tool are likewise retrievable via the SIMMER website. SIMMER's underlying chemical and protein databases will be updated whenever major releases of MetaCyc that result in new enzyme-annotated reactions are released (estimated to be once a year).

## Discussion

In this work, we created a tool that appropriately describes reaction chemistry and harnesses all current information on gut bacterial sequences, both from isolates and metagenome assembled genomes. This advances our ability to discover the genetic determinants of human microbiome chemical transformations, because previous methods for in silico metabolism prediction had several key limitations, including low accuracy. Here, we demonstrated SIMMER's ability to recover known drug-metabolizing enzymes in the human gut, performed the first species-level characterization of MTX metabolism by bacteria in the human gut microbiome, and extended previous experimental findings for multiple drug metabolism events by identifying candidate species and enzymes.

To describe chemical reactions, we were initially influenced by recent research that employed substrate and product chemistry to compare bacterial-drug metabolism events to primary reactions in the MetaCyc database, but without the end goal of EC and enzyme identity prediction (*Mallory et al., 2018*). From a reaction description standpoint, the published method was still limited in that it only included a description of one substrate and one product per reaction, precluding it from utilizing cofactors and from accurately describing anything other than intramolecular rearrangements (EC class 5, *Figure 2—figure supplement 4*). For this reason, we employed a chemical representation technique that can describe multiple inputs and outputs for a single reaction (*Schneider et al., 2015*).

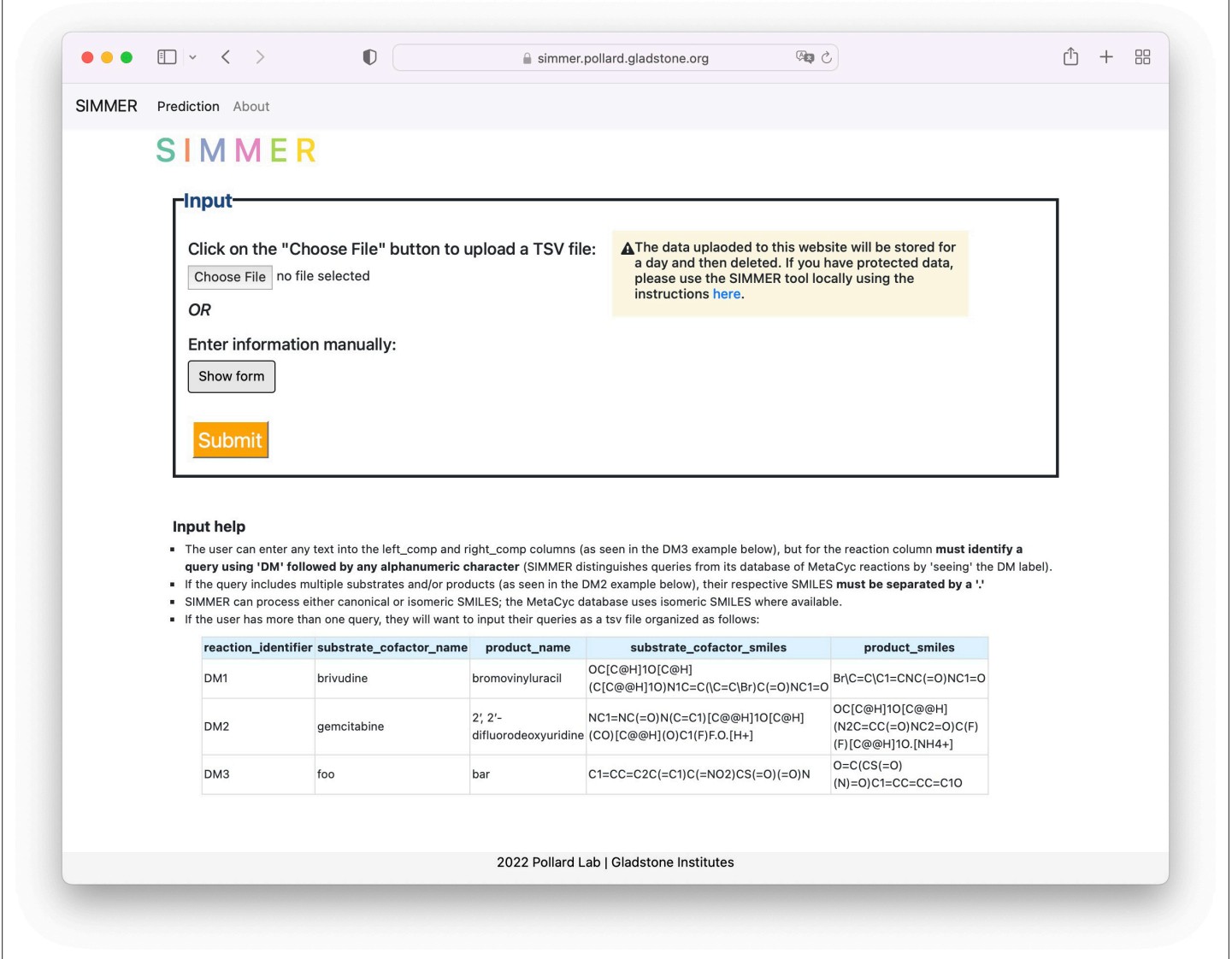

**Figure 11.** SIMMER web tool. The landing page for the SIMMER website (https://simmer.pollard.gladstone.org/) allows the user to upload a TSV file of queries or add a single query manually to run SIMMER on. It is recommended to use the command-line tool (https://github.com/aebustion/SIMMER; **Bustion, 2023**) for more than 10 input queries.

To connect these chemical descriptions to bacterial proteins in the human gut, we knew it was important not to rely on EC codes (as previous methods have done) to find relevant enzyme sequences. While EC codes are helpful for describing reaction types, they are not sufficient for predictions of microbiome orthologs due to the paucity of functional annotations in microbiome datasets (**Almeida et al., 2021**; **Thomas and Segata, 2019**). Even when present, EC entries are sometimes incomplete, with about 36% of assigned EC numbers lacking either a gene or protein sequence (**Pouliot and Karp, 2007**). Lastly, EC nomenclature fails to adequately express the complexity of peptidase enzymes (EC 3.4, about 10% of the enzymes currently classified by EC). All peptidases catalyze a nearly identical reaction, hydrolysis of a peptide bond, and as a result, the EC has lumped peptidases of diverse functions into only a few low-resolution sub-classes (**McDonald and Tipton, 2023**; **McDonald and Tipton, 2014**). For these reasons, we instead chose to create sequence searches of large genome databases directly from enzymes known to conduct chemically similar reactions, whether or not they have been fully annotated with an EC code.

When comparing database entries based on sequence similarity, the algorithm employed plays a critical role. Homology searches in pharmacomicrobiomics research are often conducted using

pairwise search algorithms such as BLAST (*Altschul et al., 1990*). A limitation of this method is that substitutions, deletions, and insertions are penalized by a set amount, regardless of where in the alignment they occur. For a given collection of functional enzymes, however, sequence conservation varies at different sites in the protein, as a result of differing strengths of selection pressures on different residues (i.e., high conservation at active sites versus low conservation in disordered domains). This position-specific information can be leveraged by performing homology searches with pHMMs, which encode protein family evolutionary patterns present in a multiple sequence alignment (*Eddy, 1998*). In the antibiotic resistance protein space, for example, pHMMs that incorporate position-specific information have found distant homologs with retained function not recovered via pairwise search methods (*Gibson et al., 2015*). pHMM searches are an improvement over BLAST from the standpoint of finding distant homologs and from the standpoint of finding targets with retained activity, as previous research has shown that global sequence identity does not necessarily map to similar function (*Gerlt et al., 2012*; *Koppel et al., 2018*). While an improvement, pHMM methods can still be uncorrelated with functional similarity and SIMMER enzyme predictions will include many false positives and require experimental validation. The goal of SIMMER, therefore, is to prioritize and limit the set of possible experiments one might do rather than to replace experiments.

SIMMER achieved high accuracy when applied to known drug metabolism events in the gut microbiome. Correct EC designations and enzyme sequences were recovered for 33 drug metabolism events previously characterized in microbiome literature. These reactions span multiple EC classes, and were described by multiple publications, demonstrating the wide application and accuracy of SIMMER. While SIMMER provides accurate (i.e., true positive) enzyme predictions for chemical transformations in the human gut, the potential for false positives may be high, as its enzyme lists are not filtered by biologically relevant metrics like substrate affinity or flux consistency in a microbial community. To the former point, users may wish to employ tools like Similarity Ensemble Approach to narrow in on hits most likely to interact with compounds of interest (*Keiser et al., 2007*). To the latter, a user could choose to further analyze their SIMMER output for flux balance if the predicted SIMMER bacterial species are described in current metabolic reconstruction models (*Heinken et al., 2020*; *Magnúsdóttir et al., 2017*).

Due to its accurate predictions of previously described drug-metabolizing bacterial enzymes, we also used SIMMER to identify previously unknown bacterial species and enzymes at play in known gut microbiome drug transformations. Recent high-throughput experimental research has greatly increased our knowledge of the number of drugs altered by bacteria in the human gut, but has led to a bottleneck in identifying the responsible bacterial enzymes. While direct experimentation is a necessary component to elucidating the bacterial players responsible, in silico methods like SIMMER are needed to help prioritize which of the many bacterial species and enzymes to assess. Here, we showed that SIMMER both corroborates previous high-throughput experimental data, and also adds increased clarity to the findings. While a previous experimental study was able to elucidate the importance of an isolate *C. scindens* in the metabolism of dexamethasone, the abundance of *C. scindens* in human samples did not correlate with metabolism. When assessed with SIMMER, however, a significant correlation between metabolite production and abundance of SIMMER enzyme predictions was observed. This finding demonstrates that species identity alone is not enough to explain bacterial chemical transformation, and that responsible genetic elements must be interrogated as well.

Lastly, we used SIMMER to perform the first species-level characterization of MTX metabolism by bacteria in the human gut microbiome. While it has been suspected for some time now that inter-individual variation in patient response to oral MTX may be due to bacterial hydrolysis of the therapeutic, no characterizations have yet been made of the species of enzymes responsible. We demonstrated the utility of using SIMMER in such a use-case. The most similar MetaCyc reaction to MTX hydrolysis, folate hydrolysis, that SIMMER identified makes much logical sense based on substrate and metabolite structures, and overall reaction chemistry. Further adding confidence to SIMMER's sequence predictions was their enrichment in the shotgun sequenced stool samples of RA patients exhibiting little response to MTX treatment. This finding plus our demonstration that human gut bacterial species are capable of depleting MTX in isolation provides provides additional support for the role of gut bacterial drug metabolism in interfering with the treatment of RA.

Two previous computational tools exist for describing non-antibiotic microbial drug metabolism. MicrobeFDT groups thousands of compounds based on their similarity to one another and annotates

compound groups based on any known links to EC numbers, and subsequently, microbes known to contain such EC codes (*Guthrie et al., 2019*). This network approach was an important addition to the exploration of microbiome metabolism, but its use is limited to a fixed database of chemicals and EC annotations which prevents the user from exploring novel chemistry and also from utilizing hypothetical protein data gathered from metagenomic sequencing studies. Furthermore, MicrobeFDT's accuracy within its database of substrates is limited by its exclusive description of substrates rather than full reactions. DrugBug, a tool that employs Random Forests rather than a network approach, also exhibits limited power and accuracy due to its sole reliance on substrate chemistry and relatively small database of only 491 isolated bacterial genomes from the human gut (*Sharma et al., 2017*). Of note, our comparison of SIMMER's performance to existing methods necessitated downloading and analyzing our positive control list against the other tools, as none of the previous publications provided any computational validation or accuracy metrics.

One user pitfall of SIMMER in comparison to previous methods, is that a reaction's product(s) and cofactor(s) identity is required to achieve the high accuracy enzyme predictions described here. This is a limitation, as a growing amount of liquid chromatography–mass spectrometry (LC–MS)/MS data in microbiome research only reports whether or not a compound is depleted in the microbiome and the mass/charge ratio of the product formed, not the product identity. While it is technically possible for the user to submit a SIMMER query that only consists of a substrate, or uses a compound identity as both substrate and product, we do not recommend this due to the previously discussed lack of accuracy when only considering substrates (*Figure 2—figure supplement 4*, *Table 1—source data 1*). For users wishing to utilize SIMMER with a compound of interest and its either unknown or uncharacterized products, additional tools such as BioTransformer could be used in tandem to create product template predictions before querying (*Djoumbou-Feunang et al., 2019*). Lastly, if the user does not hold a certain level of knowledge in chemistry, appropriate cofactors (such as water employed in a hydrolysis event) might be omitted from a query, leading to lower accuracy predictions. If a user is unsure which cofactors may be at play in their reaction of interest, reaction rules tools such as RetroRules could be employed (*Duigou et al., 2019*).

Another limitation of SIMMER is that its underlying protein data is solely metagenomics data from the human gastrointestinal tract, but some compounds, such as the vaginal gel tenofovir, are known to be altered by bacteria in non-GI tract settings (*Klatt et al., 2017*). That being said, for transformations in the human gut, SIMMER employs the largest available database of relevant bacterial sequences, and the tool could easily be expanded in the future to include other human body sites as well as non-host associated environments. Further related to database constraints, while SIMMER is novel in its ability to query reactions not previously described in chemistry databases, its search space is still limited to reactions that broadly relate to those captured in MetaCyc. While MetaCyc is the largest annotated bacterial reaction database to date (*Altman et al., 2013*; *Caspi et al., 2020*), it likely only represents a fraction of the still underexplored microbiome catalyzed biotransformation space. Additionally, the chemistry present in MetaCyc is not restricted to anaerobic organisms such as the bacteria found in the human gut. As MetaCyc expands, or additional databases get employed, SIMMER will be able to make increasingly fine-tuned predictions.

SIMMER enters microbiome biotransformation research at an important point: while there are hundreds of microbiome altered compounds which are in need of enzyme identification, there are also a sufficient number with characterized enzymes to enable us to test the tool's accuracy. Its ability to predict these known enzymes accurately builds confidence for its predictions of yet unknown enzymes. With this tool in hand, microbiome researchers can make informed hypotheses before embarking on the lengthy laboratory experiments required to characterize novel bacterial enzymes that can alter human ingested compounds. Continued refinement of SIMMER and other computational tools will accelerate microbiome research, providing data-driven hypotheses for experimental testing and a first step toward understanding the full scope of metabolism by the human microbiome.

## Materials and methods

### Key resources table

| Reagent type (species) or resource | Designation | Source or reference | Identifiers | Additional information |
|---|---|---|---|---|
| Chemical compound, drug | Methotrexate | Sigma-Aldrich | M9929-100MG | |

*Continued on next page*

| Reagent type (species) or resource | Designation | Source or reference Identifiers | Additional information |
|---|---|---|---|
| Strain, strain background (*Bacteroides thetaiotaomicron*) | *Bacteroides thetaiotaomicron* | DSM 2079 | |
| Strain, strain background (*Clostridium asparagiforme*) | *Clostridium asparagiforme* | DSM 15981 | |
| Strain, strain background (*Clostridium sporogenes*) | *Clostridium sporogenes* | ATCC 15579 | |
| Strain, strain background (*Clostridium symbiosum*) | *Clostridium symbiosum* | DSM 934 | |
| Strain, strain background (*Akkermansia muciniphila*) | *Akkermansia muciniphila* | DSM 22959 | |
| Strain, strain background (*Alistipes shahii*) | *Alistipes shahii* | DSM 19121 | |
| Strain, strain background (*Bacteroides caccae*) | *Bacteroides caccae* | DSM 19024 | |
| Strain, strain background (*Bacteroides eggerthii*) | *Bacteroides eggerthii* | DSM 20697 | |
| Strain, strain background (*Bacteroides finegoldii*) | *Bacteroides finegoldii* | DSM 17565 | |
| Strain, strain background (*Bacteroides fragilis*) | *Bacteroides fragilis* | DSM 2151 | |
| Strain, strain background (*Bacteroides ovatus*) | *Bacteroides ovatus* | DSM 1896 | |
| Strain, strain background (*Bacteroides stercoris*) | *Bacteroides stercoris* | DSM 19555 | |
| Strain, strain background (*Bacteroides uniformis*) | *Bacteroides uniformis* | DSM 6597 | |
| Strain, strain background (*Bacteroides vulgatus*) | *Bacteroides vulgatus* | DSM 1447 | |
| Strain, strain background (*Bacteroides xylanisolvens*) | *Bacteroides xylanisolvens* | DSM 18836 | |
| Strain, strain background (*Bifidobacterium longum*) | *Bifidobacterium longum subsp longum* | DSM 20219 | |
| Strain, strain background (*Blautia producta*) | *Blautia producta* | DSM 3507 | |
| Strain, strain background (*Clostridium innocuum*) | *Clostridium innocuum* | DSM 1286 | |
| Strain, strain background (*Clostridium scindens*) | *Clostridium scindens* | DSM 5676 | |
| Strain, strain background (*Collinsella aerofaciens*) | *Collinsella aerofaciens* | DSM 3979 | |
| Strain, strain background (*Dorea formicigenerans*) | *Dorea formicigenerans* | DSM 3992 | |
| Strain, strain background (*Dorea longicatena*) | *Dorea longicatena* | DSM 13814 | |
| Strain, strain background (*Edwardsiella tarda*) | *Edwardsiella tarda* | ATCC 23685 | |
| Strain, strain background (*Eggerthella lenta*) | *Eggerthella lenta 1-3-56* | DSM 110906 | |
| Strain, strain background (*Eggerthella lenta*) | *Eggerthella lenta* | DSM 2243 | |
| Strain, strain background (*Eggerthella sinensis*) | *Eggerthella sinensis* | DSM 16107 | |
| Strain, strain background (*Eggerthella sp.*) | *Eggerthella sp.* | DSM 11767 | |
| Strain, strain background (*Enterococcus faecalis*) | *Enterococcus faecalis* | DSM 2570 | |
| Strain, strain background (*Escherichia coli*) | *Escherichia coli BW25113* | DSM 27469 | |
| Strain, strain background (*Eubacterium eligens*) | *Eubacterium eligens* | DSM 3376 | |
| Strain, strain background (*Eubacterium hallii*) | *Eubacterium hallii* | DSM 3353 | |
| Strain, strain background (*Eubacterium rectale*) | *Eubacterium rectale* | DSM 17629 | |

| Reagent type (species) or resource | Designation | Source or reference Identifiers | Additional information |
|---|---|---|---|
| Strain, strain background (*Faecalibacterium prausnitzii*) | *Faecalibacterium prausnitzii* | DSM 17677 | |
| Strain, strain background (*Fusobacterium nucleatum*) | *Fusobacterium nucleatum subsp. nucleatum* | DSM 15643 | |
| Strain, strain background (*Gordonibacter pamelae*) | *Gordonibacter pamelae* | DSM 110924 | |
| Strain, strain background (*Lactococcus lactis*) | *Lactococcus lactis* | DSM 20481 | |
| Strain, strain background (*Lactonifactor longoviformis*) | *Lactonifactor longoviformis* | DSM 17459 | |
| Strain, strain background (*Parabacteroides distasonis*) | *Parabacteroides distasonis* | DSM 20701 | |
| Strain, strain background (*Parabacteroides merdae*) | *Parabacteroides merdae* | DSM 19495 | |
| Strain, strain background (*Prevotella copri*) | *Prevotella copri* | DSM 18205 | |
| Strain, strain background (*Providencia rettgeri*) | *Providencia rettgeri* | DSM 4542 | |
| Strain, strain background (*Ruminococcus obeum*) | *Ruminococcus obeum* | DSM 25238 | |

## Preparation of SIMMER's underlying chemical data

13,387 gene-annotated, enzyme-driven bacterial reactions were downloaded from MetaCyc version 24.1 (*Caspi et al., 2008*). Each reaction contained Simplified Molecular Input Line Entry System strings (SMILES) of reactant(s) and product(s) and UniProt or Entrez identifiers for sequences that catalyze the reaction (we also had access to an expanded list of sequence annotations for these reactions from a personal correspondence with Peter Karp of MetaCyc). All MetaCyc compounds were then protonated based on the pH environment of 7.4 in the human small intestine, where most oral drug absorption occurs. Protonation states were calculated using ChemAxon's cxcalc majorms software (*ChemAxon, 2023* ).

RDKit's rdChemReactions module was employed to create chemical fingerprints representing each MetaCyc reaction. Chemical reaction objects were constructed from reaction SMILES arbitrary target specification (SMARTS) strings. Fingerprints for these reactions were then created using the resulting difference of product(s) and reactant(s) Atom-Pair fingerprints (*Schneider et al., 2015*). SIMMER users can also opt to use Topological Torsion, Pattern, or RDKit fingerprints, but unless otherwise stated, all analyses in this manuscript use Atom-Pair difference fingerprints. Of the 13,387 MetaCyc reactions, 8914 were able to be fingerprinted using this method. Failed fingerprints were due to ambiguous SMILES identifiers or presence of non-small-molecule compounds in a reaction, such as peptides.

After creating fingerprint vectors for all MetaCyc reactions, an 8914 by 8914 pairwise similarity matrix of Tanimoto coefficients was created. These Tanimoto vectors make up SIMMER's underlying chemical data.

## Preparation of SIMMER's underlying protein data

For each of the 8914 fingerprinted MetaCyc reactions, all relevant gene sequences were retrieved from the MetaCyc reaction's UniProt and Entrez database linkouts and additional sequence data acquired from Peter Karp at MetaCyc. If at least two genes, with a median pairwise sequence similarity greater than or equal to 27%, were linked to a given MetaCyc reaction, the sequences were used to create a multiple sequence alignment and subsequent pHMM using Clustal Omega and HMMER3 (version 3.2.1) software, respectively (*Eddy, 2009*; *Sievers and Higgins, 2014*). This similarity cutoff was chosen based on previous protein family literature (*Mi et al., 2021*). If fewer than two genes, or genes with less than 27% global similarity, were associated with a given MetaCyc reaction, a pHMM of the MetaCyc gene(s) PANTHER subfamily was retrieved via InterPro linkouts (*Mi et al., 2021*). MetaCyc derived and PANTHER subfamily pHMMS were then queried against a UHGG collection of 286,997 isolate genomes and metagenome assembled genomes from the human gut environment using the HMMER3 hmmsearch module (*Almeida et al., 2021*; *Eddy, 2009*). In the case of MetaCyc

reactions with too few sequences, too low a median pairwise sequence identity, *and* a missing PANTHER database subfamily pHMM, single sequence protein queries were conducted against the UHGG databse using HMMER3's phmmer module, which internally created protein profiles for the single query sequences based on a position-independent scoring system. Resulting enzyme hit lists were filtered to only include high significance hits (e-value < 1E − 5, and hit length ≥ half of the input pHMM alignment or single sequence length). In sum, for each MetaCyc reaction, a profile representing the diversity of the enzyme family for that chemical transformation was used to find sequence similar hits in the human gut microbiome that can mediate chemically similar reactions.

Each human gut microbiome hit was further described by the identity, prevalence, and abundance of the bacterial strain in which it resides. To establish prevalence and abundance of UHGG strains, metagenomic analysis was performed on the Predict (Personalized Responses to Dietary Composition Trial) cohort due to its high number of samples and favorable sequencing depth (*Asnicar et al., 2021*). Shotgun metagenomic reads were analyzed with MIDAS2 an implementation of Metagenomic Intra-Species Diversity Analysis Subcommands (MIDAS) (*Nayfach et al., 2016*; *Zhao et al., 2022*) designed for the UHGG database. Presence of a SIMMER predicted species in a given sample was established when reads mapped (HS-BLASTN) to 15 single-copy universal genes for that species (*Chen et al., 2015*), with at least 75% alignment coverage. To assess the gene content of a sample, shotgun metagenomic reads were aligned to a MIDAS2 created pangenome of the SIMMER species' genes clustered at 99% nucleotide identity. Copy number of a SIMMER gene prediction was established by dividing aligned prediction reads by the full length of the prediction. This number was then normalized by the read coverage of 15 single-copy universal genes in the same sample to estimate copy number per cell. Presence of a SIMMER enzyme was established if at least 0.35 gene copies per cell were present in a sample.

Phylogenetic trees were also constructed for each hmmsearch and phmmer result. For each set of MetaCyc reaction human gut microbiome enzyme hits, CD-HIT was used to cluster results at 95% identity (*Fu et al., 2012*). Then MUSCLE was used to create a multiple sequence alignment for input to FastTree (*Edgar, 2004*; *Price et al., 2009*). Compact tree visualizations were made in R using ggtree and ggtreeExtra (*Xu et al., 2021*; *Yu et al., 2017*). All tree tips were colored by phylum and surrounded by circle annotators describing a given hit's Prokka predicted function, genome type (i.e., from an isolate or metagenome assembled genome), and prevalence/abundance in the Predict cohort (*Seemann, 2014*).

## Query functionality of SIMMER

The query functionality of SIMMER was designed similar to the precomputed chemistry data. After receiving an input chemical transformation (or tsv describing multiple input reactions) in the form of SMILES, SIMMER fingerprints the reaction(s) and compares it to the precomputed chemical space by computing the Tanimoto coefficients between the input(s) and all precomputed reactions. The 8914 precomputed MetaCyc reaction Tanimoto vectors are then sorted by ascending euclidean distance to the query Tanimoto vector. SIMMER by default ranks reactions' euclidean distances based directly on the Tanimoto vectors, but if a user's inputs require a decrease in computational burden, PCA can be employed after similarity matrix creation and before euclidean distance rankings. The number of PCs to be used depends on the fingerprint style employed and was determined by the Kaiser criterion. Unless otherwise stated, all analyses in this manuscript employed the full Tanimoto similarity matrix with no PCA reduction. Human gut microbiome enzymes that may conduct the input reaction are reported from the precomputed UHGG hmmsearch or phmmer results of the closest euclidean distance MetaCyc reaction. Significantly enriched EC identities (i.e., reaction types) are also reported.

## Reaction type predictions

SIMMER predicts an EC code (i.e., reaction type) for a query reaction if there is an enrichment of a particular EC at the top of the reaction list. Enrichment was determined in a manner similar to GSEA (*Subramanian et al., 2005*). For each EC code associated with MetaCyc reactions, an enrichment score (ES) was calculated by walking down the ranked list of reactions. Starting with a score of zero, each time the given EC is encountered the score increases by one, and each time a different EC is encountered the score decreases by one. At the end of this process, each EC receives an ES that is the score's maximum distance from zero after walking through the list (*Figure 2—figure supplement 1A*).

Because the MetaCyc database of reactions is unbalanced in its EC code representation, ES scores for a given EC type are divided by the number of times the EC in question occurs in the database. This yields a normalized ES (NES) for SIMMER reporting. Significance is established by comparing the true NES to the NES achieved from 1000 permutations of a shuffled reaction list (*Figure 2—figure supplement 1B, C*). When multiple EC codes are predicted as significant, they are ranked in ascending order of where in the list of 8914 reactions the NES occurs. This method was verified by subsampling the database of MetaCyc reactions to equal numbers (*N* = 96) of reactions for each EC class, the broadest resolution level of an EC code. Each of these subsampled reactions was then queried with SIMMER against the entire MetaCyc reaction database to create sorted reaction lists for each query. SIMMER predicted an EC code(s) for each reaction based on the most highly enriched EC. SIMMER's recall, precision, and accuracy are high for EC class, sub-class, and sub-sub-class level resolution (*Figure 2B*, *Figure 2—source data 1*). For the serial designation of an EC code (the most granular description of an EC code), however, SIMMER's performance diminished, potentially because enrichment calculations suffer from increased uniqueness in the ranked list and therefore reduced power to determine a match (*Figure 2—source data 1*). This indeed appears to be the case; when the database is subsampled to ensure at least three of each unique serial designation, *F*1-scores (the harmonic mean of precision and recall) and accuracy remain high despite the increased EC resolution (*Figure 2—source data 1*, *Figure 2—figure supplement 1D*).

## Euclidean distance silhouette scores

To analyze SIMMER's resilience to different reaction chemistry representations, we created a silhouette-like euclidean distance score. For the precomputed MetaCyc chemical dataset of 8914 reactions (i.e., the Tanimoto pairwise similarity matrix), we split all reactions into their top-level EC codes (i.e., EC class) and determined for each reaction its euclidean distance to all reactions within its EC class versus outside its EC class. From the two distributions (within EC and without EC distances) created, we computed a Kolmogorov statistic to determine if the distributions significantly ($p < 0.05$) differed. We repeated this process for finer resolution EC classifications (sub-class, sub-sub-class, and serial designation). Euclidean distance silhouette scores were used to compare different chemical representations, such as fingerprint style, inclusion of products, and inclusion of cofactors.

## Relationship between SIMMER's underlying chemical and protein data

For MetaCyc enzymes (*N* = 34,279) associated with multiple reactions, one reaction was used as a SIMMER query, and the other reaction(s) searched for in the ordered reaction list output. As a negative control, these reaction similarity results were then compared to all pairwise combinations of MetaCyc enzymes (subsampled to *N* = 34,279) that do not conduct the same reaction.

We also assessed the relationship between chemistry and protein similarity for all pairwise combinations of a subset of MetaCyc reactions annotated with only one protein sequence (*N* = 604 reactions). Chemical similarity was based on the Euclidean distance between two reaction fingerprint vectors in SIMMER's precomputed chemical space (*Figure 1A*). Global protein similarity was determined via the Needleman–Wunsch algorithm. The relationship between chemical similarity and protein similarity was assessed with a Pearson's correlation coefficient and p-value calculated using a Wald test with *t*-distribution of the test statistic.

## Creating a compendium of drug metabolism use cases from the human gut

To analyze SIMMER under the use-case of drug metabolism, we created a compendium of drug degradations that occur in the human gut microbiome. The compendium of reactions is based on a literature curation of hundreds of papers, and is organized by reactions producing known/unknown metabolites and driven by known/unknown bacterial enzymes. The drug metabolism positive controls used to assess SIMMER's accuracy were drawn from the list of reactions possessing a structurally elucidated metabolite and driven by a characterized bacterial enzyme.

We further expanded the positive control list to include sequence similar enzymes that likely perform the same function. For this expansion, we performed pHMM searches (when a positive control reaction had been characterized with multiple sequence similar enzymes) and phmmer searches (when a positive control reaction had been characterized with only one sequence) of the UHGG database

using HMMER3 software (*Almeida et al., 2021*; *Eddy, 2009*). High significance (e-value < 1E − 5) hits were kept when the resulting alignment was at least 50% of the input pHMM or sequence length. This list of significant hits was filtered by presence in human ileum or jejunum (the site of human drug absorption) via DIAMOND searches against metagenomic reads from a published study that employed jejunum and ileum endoscopy (*Buchfink et al., 2021*; *Zmora et al., 2018*). The hits were also filtered for presence in RNAsequencing data via DIAMOND searches of rnaSPAdes assembled reads from HMP2 metatranscriptomics control patient samples (*Bushmanova et al., 2019*; *Integrative HMP (iHMP) Research Network Consortium, 2019*). The hits were lastly filtered by predicted affinity for their substrates using the Similarity Ensemble Approach (*Keiser et al., 2007*).

## Comparison to existing methods

DrugBug predictions were made using the DrugBug web tool (http://metagenomics.iiserb.ac.in/drugbug/) with default settings. MicrobeFDT predictions were made in two manners: first by looking for direct enzyme metabolism events, and second, by looking for enzyme metabolism of compounds that overlap chemically with the positive control in question. Cypher query commands used are included in *Table 1—source data 1*.

## Analysis of 16S and metagenomics data from Javdan et al.

We analyzed results from sequencing studies (NCBI BioProject: PRJNA593062) described in a previously published high-throughput investigation of bacterial drug metabolism in human stool samples (*Javdan et al., 2020*). The first sequencing set in this publication was a deep metagenomic sequencing of one pilot individual's ex vivo stool originally evaluated for its ability to degrade hundreds of therapeutics. We used MetaSPAdes with default settings to assemble the metagenomics reads into scaffolds (*Nurk et al., 2017*). We then queried SIMMER with eight reactions that were structurally elucidated (via nuclear magnetic resonance) by the previous publication, and ensured via TBLASTN searches that SIMMER predicted hits were found in the assembled metagenomic reads. The second sequencing set was a 16S rRNA sequencing experiment of twenty human donor stool samples originally evaluated for their inter-individual variation in bacterial drug degradation. We queried SIMMER with five of these reactions possessing structurally elucidated metabolites, and evaluated enrichment of SIMMER predicted bacterial species in metabolizing versus non-metabolizing donors. Species matches between SIMMER species predictions and the 16S study were made using the SequenceMatcher class from the difflib python module set to an 80% ratio cutoff. Enrichment of SIMMER predicted bacterial genomes was then assessed by computing a *t*-test for number of SIMMER genomes in metabolizers versus number of SIMMER genomes in non-metabolizers for a given reaction.

## Analysis of dexamethasone metagenomics data

For experimental corroboration of dexamethasone metabolism, we accessed shotgun sequencing data (PRJEB31790) from a cohort of 28 human stool samples shown to metabolize dexamethasone to varying degrees (*Zimmermann et al., 2019b*). Shotgun reads were assembled using MetaSpades with default settings. The metabolism of dexamethasone to 17-oxodexamethasone was input to SIMMER with $N = 20$ reactions output, and a DIAMOND database of all SIMMER enzyme predictions from *C. scindens* was created. Presence of SIMMER enzyme predictions was established via search with DIAMOND and normalized by sample read depth. Significance was established with a Pearson's correlation coefficient and p-value calculated using a Student's *t*-distribution.

## Analysis of RA metagenomics data

The Artacho sequencing study (PRJNA682730) raw reads were assembled using MetaSpades with default settings (*Nurk et al., 2017*). After assembly, DIAMOND was used to search for SIMMER sequences in reads, with presence defined as at least 50% coverage and at least 97% identity. All abundance measures were normalized by read depth. Correlation between DAS28 improvement and number of aligned SIMMER enzyme predictions was assessed using Pearson's correlation coefficient and p-value calculated using a Student's *t*-distribution. Enrichment of SIMMER enzyme predictions in MTX responders versus non-responders was assessed using a generalized linear model, glm(count~response, family = poisson).

## Bacterial isolate screen for MTX hydrolysis

42 isolates commonly found in the human gut microbiome were obtained from Deutsche Sammlung von Mikroorganismen und Zellkulturen (DSMZ) and American Type Culture Collection (ATCC), and subcultured as previously described (*Nayak et al., 2021*). MTX (100 μg/ml) was added to cultures for 72 hr, samples then spun down at 2000 rcf for 5–10 min at 4°C, and supernatant injected to HPLC (see HPLC method). MTX was dosed based on the predicted concentration of the drug in a human gastro-intestinal tract, as previously described (*Nayak et al., 2021*). Drug depletion was defined as at least a 50% decrease in MTX levels compared to control. SIMMER predicted enzymes' presence or absence in the 42 isolates was determined by downloading genomes (*Figure 8—source data 1*) for all 42 isolates and conducting DIAMOND searches. Presence was defined as at least 97% global percent identity. Presence of AcrAB efflux machinery was determined by conducting DIAMOND searches for *E. coli* AcrA (P0AE06) and AcrB (P31224) against all 42 isolates, and reported in *Figure 8—source data 1*. For the phylogenetic tree, 16S rRNA gene sequences for the 42 organisms were downloaded from the Greengenes database (*DeSantis et al., 2006*) and aligned using MUSCLE (*Edgar, 2004*). Gaps occurring in greater than 50% of sequences removed before creating a maximum likelihood phylogenetic tree (PhyML) with 100 bootstraps and the GTR substitution model (*Guindon et al., 2010*).

## MTX HPLC method

HPLC assays were performed on an Agilent HPLC (1220 Infinity), and data collected with OpenLAB CDS (Agilent Technologies). Solvent A was 0.1% formic acid, and solvent B was 100% methanol. Solvent B concentration was 10–30% from 0 to 1 min, 30–100% from 1 to 7 min, and then 100–10% from 7 to 7.5 min. The flow rate was 0.6 ml/min. A C18 column (Kinetex 2.6 μM; 100 Å; 15 cm × 0.46 cm; Phenomenex; 00F-4462-E0) was used with a SecurityGuard ULTRA cartridge guard column (Phenomenex part number AJ0-8768). The injection volume was 30 μl. At 310 nm, MTX retention time was 5.5 min (*Figure 8—figure supplement 2*). We compared the amount of MTX present in the bacterial supernatant compared to sterile and dimethyl sulfoxide (DMSO) controls to assess MTX metabolism.

## Web tool creation

We used the python web framework Flask (https://flask.palletsprojects.com/en/2.1.x/) to make SIMMER available as a user-friendly website. The website accepts either a single query reaction or multiple query reactions via a file upload and provides the same outputs as the SIMMER command-line tool. The website also allows the user to download the outputs of interest. Keeping in mind the privacy and security of the data that a user might upload to the website, the website is designed to delete all uploaded data within 24 hr from the server. This will ensure security of the uploaded data.

## Acknowledgements

We thank Russ Altman and Emily Mallory for their work that first inspired the comparison of MetaCyc reactions to known gut microbiome chemistry (*Mallory et al., 2018*), and Russ Altman for further insightful conversation. We thank Peter Karp and Peter Midford of MetaCyc for providing us with an expanded set of links between MetaCyc reactions and Uniprot or Entrez sequences. We thank Ben Guthrie of the Turnbaugh lab for collaboration and guidance in the ongoing experimental validation of SIMMER enzyme predictions. We thank Daniela Arce and Xiaofan Jin of the Pollard lab for guidance on assessing enzyme presence in the jejunum and ileum of the human intestinal tract. We thank Chunyu Zhao and Jason Shi of the Pollard lab for guidance on the appropriate use of MIDAS2 and the UHGG database. We thank Patrick Bradley for guidance on the use of HMMER3 software in a microbiome drug metabolism context. We thank members of the Pollard and Turnbaugh laboratories for their suggestions during the research process and writing of this manuscript. Annamarie Bustion was supported by a trainee pilot award from the UCSF Benioff Center for Microbiome Medicine, a predoctoral fellowship in informatics from the PhRMA foundation, an Achievement Rewards for College Scientists (ARCS) Scholarship from the ARCS foundation, and Training Grants from the NIGMS of NIH (5T32GM007175-42 and 2T32GM007175-41). This work was also supported by Gladstone Institutes. Renuka Nayak was supported by K08AR073930, Arthritis National Research Foundation, R01AR074500, Russell Engelman Rheumatology Research

Center, UCSF Perstein Award, and a UCSF Bechtel Award. PJT was supported by the National Institutes of Health (R01HL122593, R01AR074500). PJT and KP are Chan Zuckerberg Biohub-San Francisco Investigators.

## Additional information

### Competing interests

Peter J Turnbaugh: Reviewing editor, *eLife*. Katherine S Pollard: is a consultant for Phylagen Inc. The other authors declare that no competing interests exist.

### Funding

| Funder | Grant reference number | Author |
|---|---|---|
| PhRMA Foundation | Predoctoral Fellowship | Annamarie E Bustion |
| ARCS Foundation | Graduate Student Scholarship | Annamarie E Bustion |
| UCSF Benioff Center for Microbiome Medicine | Trainee Pilot Award | Annamarie E Bustion |
| Gladstone Institutes | | Katherine S Pollard |
| Chan Zuckerberg Biohub San Francisco | | Katherine S Pollard Peter J Turnbaugh |
| National Institute of General Medical Sciences | 5T32GM007175-42 | Annamarie E Bustion |
| National Institute of General Medical Sciences | 2T32GM007175-41 | Annamarie E Bustion |
| National Heart, Lung, and Blood Institute | R01HL122593 | Peter J Turnbaugh |
| National Institute of Arthritis and Musculoskeletal and Skin Diseases | R01AR074500 | Peter J Turnbaugh Renuka R Nayak |
| National Institute of Arthritis and Musculoskeletal and Skin Diseases | K08AR073930 | Renuka R Nayak |
| Arthritis National Research Foundation | | Renuka R Nayak |
| Russell Engelman Rheumatology Research Center | | Renuka R Nayak |
| University of California, San Francisco | Perstein Award | Renuka R Nayak |
| University of California, San Francisco | Bechtel Award | Renuka R Nayak |

The funders had no role in study design, data collection, and interpretation, or the decision to submit the work for publication.

### Author contributions

Annamarie E Bustion, Conceptualization, Data curation, Software, Formal analysis, Validation, Investigation, Visualization, Methodology, Writing - original draft, Writing – review and editing; Renuka R Nayak, Investigation, Writing – review and editing; Ayushi Agrawal, Resources, Software, Visualization, Writing – review and editing; Peter J Turnbaugh, Conceptualization, Resources, Supervision, Writing – review and editing; Katherine S Pollard, Conceptualization, Resources, Supervision, Funding acquisition, Project administration, Writing – review and editing

## Author ORCIDs

Annamarie E Bustion (ID) http://orcid.org/0000-0002-7380-3619
Ayushi Agrawal (ID) http://orcid.org/0000-0003-2940-8926
Peter J Turnbaugh (ID) http://orcid.org/0000-0002-0888-2875
Katherine S Pollard (ID) http://orcid.org/0000-0002-9870-6196

## Decision letter and Author response

Decision letter https://doi.org/10.7554/eLife.82401.sa1
Author response https://doi.org/10.7554/eLife.82401.sa2

## Additional files

### Supplementary files

- Supplementary file 1. Literature curated list of drug metabolism events in the human gut microbiome.
- Supplementary file 2. SMILES used as SIMMER input.
- MDAR checklist

### Data availability

Data generated and analyzed during this study are provided in Figures 2–10 source data files, Table 1 source data file, supplementary files, and at https://github.com/aebustion/SIMMER (copy archived at *Bustion, 2023*). Accession numbers of previously published datasets are provided in Materials and Methods section. SIMMER code can either be run at the SIMMER website (https://simmer.pollard.gladstone.org/) or downloaded directly from the above-linked GitHub.

The following previously published datasets were used:

| Author(s) | Year | Dataset title | Dataset URL | Database and Identifier |
|---|---|---|---|---|
| Artacho, et al | 2020 | human gut microbiome predicts response to MTX treatment | https://www.ncbi.nlm.nih.gov/bioproject/?term=PRJNA682730 | NCBI BioProject, PRJNA682730 |
| Zimmermann, et al | 2019 | Drug metabolism by 28 human gut communities | https://www.ncbi.nlm.nih.gov/bioproject/?term=PRJEB31790 | NCBI BioProject, PRJEB31790 |
| Zimmermann, et al | 2019 | Sequencing data for "Mapping drug metabolism by the human gut microbiome using personalized microbial communities" | https://www.ncbi.nlm.nih.gov/bioproject/?term=PRJNA593062 | NCBI BioProject, PRJNA593062 |

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
