## [Editor Report]

This paper provides important advances in utilizing chemical, metagenomic and enzyme mechanistic insights into the roles gut microbiota play in health-related chemical conversions. The authors convey results from a series of convincing studies that outline the utility of their computational platform, one that will be useful to both specialized microbiome researchers as well as a broad audience of scientists interested in the numerous ways non-host enzymes impact host biology.

---

## [Decision Letter]

**Decision letter after peer review:**

Thank you for submitting your article "A novel *in silico* method employs chemical and protein similarity algorithms to accurately identify chemical transformations in the human gut microbiome" for consideration by *eLife*. Your article has been reviewed by 3 peer reviewers, one of whom is a member of our Board of Reviewing Editors, and the evaluation has been overseen by Wendy Garrett as the Senior Editor. The following individual involved in the review of your submission has agreed to reveal their identity: Michael Zimmermann (Reviewer #2).

Essential revisions:

1) Additional non-experimental validations must be performed to support the conclusions drawn by the manuscript

2) At least one experimental validation must also be provided to establish the usefulness of this tool.

3) The manuscript frequently claims superiority over existing methods, but this is not supported by the necessary additional validations requested above.

*Reviewer #1 (Recommendations for the authors):*

The following should be addressed prior to consideration for publication.

1. The manuscript is far too repetitive in terms of its goals and its claims of success. These claims are clear from the Abstract and Introduction, and should not be revisited throughout the text.

2. More information should be given about the granularity present in the EC code (as noted first in the text on line 85) toward the goals stated. Are EC codes sufficient to select enzyme orthologs within an overall class? For example, would specific Azoreductases be selected by this tool (e.g., a Class I vs. Class II), or would AzoR be the output? This is important for several reasons. First, EC classifications are typically broad (e.g., glycoside hydrolase family 1 has many thousand orthologs in the gut microbiome), and within such classifications many, many distinct orthologs are present. The authors should be clear about what their tool produces and, perhaps even more importantly, what it does NOT produce.

3. How did MetaCyc filter from its ~18K reactions to the nearly ~9K employed?

4. What are the limitations of MetaCyc? Even 9K sounds like a lot, but given ~10 million unique proteins in a conservative estimation of the gut microbiome, those factors (even if only 10% are enzymes) are likely to catalyze more than 10,000 reactions.

5. The statement on lines 214-5 that the data in Figure 3B that "there is much variation in this relationship" appears to be a massive understatement. Figure 3B appears to contain much more information than the authors are considering. To name just one, what is the nature of the cluster between 0.2-0.4 on the x-axis and ~0.2 on the y-axis?

6. On line 281, an output from SIMMER is stated to provide a list of the "top 20" ranked MetaCyc reactions. Is this good enough? The manuscript claims that this tool will provide information that will allow researchers to focus their hypotheses, but if the "right answer" is only 5% of the "possible answers" provided, that is pretty unfocused and does not support the (highly, highly) repeated claims of clean deliverables proffered by the authors.

7. There appears to be package version conflicts for the majority of the conda dependencies specified in SIMMER.yml, such that implementing command-line SIMMER is not straightforward as described in the manuscript or on the github page. This needs to be corrected and β-tested so users can successfully use the tool locally in a command-line manner.

8. An additional "real world" example should be explored to examine the success of this tool. Two suggestions are PMID 31663730 or the very recently published PMID 35953888. In both cases, cleavage of glucuronides is examined from human drugs. However, very unique and hard-to-predict types of glucuronidases are identified in each case, and these unique orthologs are due to the distinct linkage of the glucuronide to the parent drug molecule. Would SIMMER be able to identify these orthologs?

*Reviewer #2 (Recommendations for the authors):*

A few suggestions to improve and clarify the manuscripts.

1) Related to comment 1) in the 'public review': presenting how many false positive hits per prediction are generated by SIMMER, belonging to how many EC classes, sub-classes, etc. together with the truly positive results would benefit the interpretation of results. Also, the rank position of the true positive prediction in the prediction list should be reported, and not only its presence, since only the top enzyme annotations are used as queries for the protein similarity search of potential gut bacterial enzymes. Especially the potential impact of the presence of several EC classes (eventually biasing the hypothesis generation process before experimental validation) should be assessed.

2) Related to comment 2) in the 'public review': due to the different input formats between SIMMER, MicrobeFDT, and DrugBug, it would be useful to demonstrate that SIMMER has at least comparable performances to the other tools when their input format is used (i.e., substrate fingerprint only).

3) Related to comment 5) in the 'public review': a revision of the text is needed to clarify terms and avoid the readers' confusion. Already in the title, the method is described as being able to "identify chemical transformations in the human gut microbiome"; however, SIMMER requires a fully chemically characterized biotransformation (in terms of substrate, (co-factors), and products) as input (and is therefore not able to identify new biotransformations). The text should be modified to highlight that SIMMER identifies and characterizes the enzymatic reaction of known biotransformation. On the same note, please also revise lines 81-82, 358-359, 480-481, and 531-532 accordingly.

4) The website version of SIMMER could be easily tested; it's intuitive, although a more extensive description of Output headers could help the user understand the results better. It was not possible to create the conda env after cloning the github repo ("Solving environment: failed" error)

5) As showcased in Figure4—figure supplement 1, additional filtering steps might be needed to filter the SIMMER enzyme list to a relevant subset (e.g., before in vitro validation). The authors should consider adding a plug-in to both the command line and online version of the tool so that these filtering steps could potentially be applied by the users as well. This might benefit especially experimentalist users, who might not be comfortable with running several additional filtering steps via cli.

6) More detailed results might improve the understanding of the reported examples. E.g., it is unclear how SIMMER could not accurately predict EC number for brivudine transformation (Line 197), but correctly predict BT_4554 as being responsible for the reaction. Reporting SIMMER output for this query as supplementary material would help the user navigate the examples better.

7) In the example describing dexamethasone biotransformation, the authors should describe the SIMMER predicted enzyme(s) in more detail (predicted EC number, sequence, etc.) since it (they) reportedly does not correspond to the already described DesAB gene.

8) When quantification of SIMMER predicted enzymes is performed in metagenome samples (see for example Lines 427, 430, 545), it is unclear what association is performed: is this the quantification of only the first (i.e., top scoring) predicted enzyme in the sample or is it the quantification of several different SIMMER enzymes in the same sample? If the latter, how are quantification results from different enzymes collapsed together?

*Reviewer #3 (Recommendations for the authors):*

I am most concerned about major overstatements related to the performance and validation of SIMMER resource predictions.

"We show that SIMMER predicts the chemistry and responsible species and enzymes for a queried reaction with high accuracy."

The validation of SIMMER predicted enzyme-drug interactions utilizes metagenomic data from a single donor stool sample in which drug metabolism was demonstrated. Other samples used as 'validation' have only 16S rRNA gene sequencing.

Their analysis, identifying microbial taxa via 16S in fecal samples that demonstrated metabolism of a drug does not support the claim that SIMMER "predicts the chemistry and responsible species and enzymes". There is no experimental or computational enzymatic validation; rather the authors infer from taxonomic representation in 16S data that the enzymes they predict metabolize the drug must be present in the bacteria found in the drug-metabolizing fecal samples. This validation is several steps removed from identifying "responsible species and enzymes". This sentence in the abstract is thus a major overstatement of their results.

"Bacterial species containing these enzymes are enriched within human donor stool samples that metabolize the query compound."

To discuss this point further; that bacterial species purported to contain specific enzymes are enriched in stool samples that metabolize a compound is insufficient to validate that SIMMER has identified microbiome enzymes that metabolize a drug. Saying that you see species increase in abundance that you think have enzymes that can metabolize a drug is far removed from their claim that they can identify specific microbiome enzyme-drug interactions.

In a later discussion of microbiome metabolism of dexamethasone, the authors state: "These results indicate that species level information alone is not enough to predict chemical transformations in a microbiome sample, but with SIMMER, knowledge of responsible enzymes can recapitulate a sample's potential for therapeutic degradation." However, using species-level information alone is precisely what they did in their 16S-based validation set. This point needs to be clarified.

Finally, they refer to these entirely computational validation approaches as "experimental validation" of SIMMER, which is inaccurate. Based on *eLife* precedent, I strongly advise experimental validation of at least one of their predicted novel enzyme-drug interactions.

Regarding sequence similarity/chemical similarity, a foundational component of the resource:

"SIMMER was created with the assumption that chemically similar reactions are mediated by sequence similar enzymes"

It would be helpful for the authors to consider the work of the Babbitt, Gerlt, Almo, and Jacobsen labs:

https://www.ncbi.nlm.nih.gov/pmc/articles/PMC3249080/

"From a survey of structurally characterized superfamilies, almost 40% are functionally diverse, i.e. different members catalyze reactions with different EC numbers (4). Thus, trivial annotation transfer by sequence homology is often not sufficient to assign function."

The authors' assertion that enzyme sequence similarity is sufficient to identify chemically similar reactions is also not well supported by experimental studies on enzyme families that metabolize drugs in the human gut.

In studies of Cgr2, a member of a microbiome drug-metabolizing enzyme family that was deeply characterized by the Turnbaugh and Balskus labs, sequence-based clustering was unable to resolve biochemical functioning.

https://elifesciences.org/articles/33953

"At all thresholds at which Cgr2 remained connected to other protein sequences, all characterized enzymes within the SSN were co-clustered, precluding the resolution of unique biochemical functions at this cutoff. "

In Figure 3B one can see the issues with protein percent identity as a metric for functional similarity. The use of a linear model is likely not appropriate here given the substantial variability in the relationship between reaction distance and percent identity. However, it is clear that the statement "reactions with similar chemistry are conducted by sequence similar enzymes" is not upheld in much of their data. As an example, in the range of identity that they are using (27% identity, per the methods) there is a large spread in reaction distances.

These nuances in sequence-function relationships that can confound microbiome drug metabolism studies are critical to deal with in such analyses and they are one reason that other resources that seek to identify microbiome-drug interactions utilize a variety of sources of evidence to assess the likelihood of specific microbiome-drug interactions.

As the authors note, each of these current tools to predict microbiome-drug interactions has limitations. For SIMMER, the major limitation, which is only briefly alluded to, is the need for a user to already know both substrate and product. As the authors note, publications experimentally characterizing microbiome-drug interactions often utilize parent compound loss as the marker for microbiome drug metabolism.

Knowing and providing the substrates and products requires chemical knowledge that could also be used to identify the type of reaction that is being carried out. This makes the SIMMER resource harder to use for users without that chemical knowledge, which is where the other resources that the authors use as a comparison (BugDrug, MicrobeFDT) may be more helpful.

Additional comments

The authors should indicate how many different modifications are represented in the 88 reactions that were queried to evaluate how well the resource works at predicting novel drug-metabolizing enzymes.

It is unclear if the SIMMER resource covers greater chemical space than the other resources. Database size does not necessarily correlate with chemical diversity. An explicit comparison of chemical space should be quantified and compared across all resources.

The authors do not represent the goals of the other resources that they use as comparisons accurately; these resources take different approaches to identify microbiome-drug predictions. As an example, the "Direct query" of the MicrobeFDT resource yielded 3/4 "correct" predictions, why does Table 1 not include this accuracy metric?

It would be helpful for the authors to discuss the limitations of MetaCyc and the representation of enzymes involves in anaerobic degradation reactions that would be expected in the gut.

The authors need to make available and searchable their data connecting microbiome taxa, enzymes and predicted drug metabolism, this would greatly broaden the utility of the resource to the community.

[Editors' note: further revisions were suggested prior to acceptance, as described below.]

Thank you for resubmitting your work entitled "SIMMER employs similarity algorithms to accurately identify human gut microbiome species and enzymes capable of known chemical transformations" for further consideration by *eLife*. Your revised article has been evaluated by Wendy Garrett (Senior Editor) and a Reviewing Editor.

The manuscript has been improved but there are some remaining issues that need to be addressed, as outlined below.

*Reviewer #3 (Recommendations for the authors):*

I am satisfied that the authors have carried out key validations of the SIMMER tool, including experimental characterization of a SIMMER prediction, the metabolism of methotrexate by hydrolases.

I ask that the authors credit prior research demonstrating that methotrexate is metabolized by hydrolases (https://www.ncbi.nlm.nih.gov/pmc/articles/PMC5082436/). They extend prior work on methotrexate here by showing strain-level characterization of MTX metabolism, a valuable addition to our understanding of drug metabolism by gut microbes.

---

## [Author Response]

Essential revisions:1) Additional non-experimental validations must be performed to support the conclusions drawn by the manuscript

In addition to our previous validations, the resubmission now includes:

An additional validation on an external dataset (Rheumatoid Arthritis patients’ clinical response to methotrexate) described in Figure 9 and lines 507–530.Two additions to the positive control set (regorafenib-glucuronide and mycophenolate-glucuronide), thanks to added knowledge from Reviewer 1. This expanded our list of enzyme-characterized microbiome biotransformations from 31 to 33, and we have adjusted the accuracy measures in Table 1, Figure 5, and the text accordingly.An assessment of how the accuracy and false positive rate of SIMMER’s enzyme output relate as a user probes the N (now a user defined parameter) closest reactions to a query (described in Figure 5—figure supplement 2 and lines 295–302). We additionally detail (in Figure 5B and in lines 283–286) our rationale for using N=20 reactions when calculating SIMMER’s enzyme prediction accuracy.A false positive rate for SIMMER’s EC predictor, described in Figure 2— figure supplement 2.Additional comparisons to the two existing methods. We previously described competing methods and their abilities to predict EC numbers and enzyme sequences in comparison to SIMMER (Table 1). We have supplemented the text to clarify how this analysis was conducted (lines 979– 985), and we now provide further demonstration (via modifying SIMMER to use only substrates as inputs) that SIMMER’s improved accuracy is in part due to its chemical representation method (described in lines 369–384, Table 1—source data, Figure 2—figure supplement 4). We also applied the existing methods to data from our experimental validation of bacterial metabolism of methotrexate and compared the results to SIMMER’s predictions (described next in Essential Revision 2).

2) At least one experimental validation must also be provided to establish the usefulness of this tool.

In addition to SIMMER’s ability to predict strains and enzymes that explain interindividual metabolism variation in previously published studies (i.e., validation on external datasets), we now include a novel experimental assessment of bacterial strains SIMMER predicted as capable of degrading methotrexate (MTX), the mainline therapy for rheumatoid arthritis patients. We observed a significant concordance between SIMMER predictions and depletion of MTX measured with high-performance liquid chromatography (HPLC) (OR=5.4, Fisher’s Exact Test pvalue<0.05). This experimental validation is described in a new figure (Figure 8 and lines 469–487 of the current manuscript), and represents the first strain-level characterization of MTX metabolism by human gut microbiome bacteria.

As an additional comparison to DrugBug and MicrobeFDT, we assessed MTX metabolism with the two tools. Both incorrectly predicted that MTX degradation is through transferases rather than hydrolases. Because DrugBug also provides species and enzyme predictions, we also assessed its ability to predict species responsible for MTX metabolism and found that only SIMMER was able to significantly predict the depleting species (SIMMER Odds Ratio=5.4 versus DrugBug Odds Ratio=0, Figure 8— figure supplement 1 and lines 489–497). DrugBug produced zero true positives and predicted that MTX-metabolism was a feature largely restricted to members of Bacteroidetes, a group who exhibited no activity in our *in-vitro* assay (Figure 8—figure supplement 1).

3) The manuscript frequently claims superiority over existing methods, but this is not supported by the necessary additional validations requested above.

To address this recommendation, we revised the manuscript in multiple places to temper overstatements of superiority. We also edited and supplemented the text to make clear the method comparison previously conducted. We updated accuracy measures of all three methods to reflect the new additions to the positive control list (Table 1). We provided additional demonstration, via modifying SIMMER to use only a substrate as input, that DrugBug and MicrobeFDT exhibit lower accuracy predictions due to their limited chemical representations (described in lines 369– 384, Table 1—source data, Figure 2—figure supplement 4). Finally, we performed additional comparisons to these two competing methods by evaluating their predictions for MTX metabolism in our in vitro experiments, and observed lower accuracy (described in Response 2 above).

Reviewer #1 (Recommendations for the authors):The following should be addressed prior to consideration for publication.1. The manuscript is far too repetitive in terms of its goals and its claims of success. These claims are clear from the Abstract and Introduction, and should not be revisited throughout the text.

We have revised the manuscript extensively to remove repetitive statements.

2. More information should be given about the granularity present in the EC code (as noted first in the text on line 85) toward the goals stated. Are EC codes sufficient to select enzyme orthologs within an overall class? For example, would specific Azoreductases be selected by this tool (e.g., a Class I vs. Class II), or would AzoR be the output? This is important for several reasons. First, EC classifications are typically broad (e.g., glycoside hydrolase family 1 has many thousand orthologs in the gut microbiome), and within such classifications many, many distinct orthologs are present. The authors should be clear about what their tool produces and, perhaps even more importantly, what it does NOT produce.

We have now added a quick description of what EC codes represent, in terms of increasing chemical granularity across the four-digit identifier in lines 166–172.

We thank the reviewers for highlighting that we did not adequately describe how SIMMER conducts protein similarity searches. EC predictions are not the primary output of SIMMER, and they are not the basis for SIMMER’s enzyme ortholog searches. SIMMER’s enzyme sequence and EC predictors are two separate functions that do not rely on each other. From the most similar MetaCyc reaction(s), we use MetaCyc enzyme *sequence* annotations to perform profile hidden Markov model (pHMM) searches against the UHGG database to find microbiome homologs of the annotated enzymes. These sequence searches do not use MetaCyc EC numbers in any way. As an *auxiliary* function, SIMMER also predicts EC numbers for input reactions through a separate process inspired by gene set enrichment analysis (details in Methods lines 893–923 and Figure 2—figure supplement 1). We added EC prediction since this is the primary output of existing tools and can provide useful information beyond specific enzyme hits.

To address this important point, we reworded the Results (lines 158–166) and Discussion (lines 640–654) of the text, and edited Figure 1 to make it explicit that these are two separate functions of the tool.

All this being said, the question still remains whether or not specific classes of a given enzyme can be drawn out using SIMMER’s pHMM sequence searching method. In short, it depends on the input reaction in question and how many examples of similar chemistry exist in the reaction database (MetaCyc). Enzyme class specificity of SIMMER will improve as databases continue to grow. This is possible with a method like SIMMER precisely because we do not rely on EC annotations; as MetaCyc is updated with new sequence annotated chemistry, SIMMER results will reflect orthologs based on sequence rather than database EC annotations that are missing ~70% of the time (Pouliot and Karp, 2007).

To conclude, we very much agree with the reviewers that EC codes are not sufficient for functional predictions of orthologs, and this was a key point that informed SIMMER’s architecture. By using pHMM searches for the most similar enzyme sequences and returning specific enzyme sequences, SIMMER provides a more specific output than would be possible with EC predictions alone. Thanks to the reviewers’ helpful feedback, we believe the text reads more clearly now, and readers will be able to determine from the text and from Figure 1 how SIMMER treats sequence predictions differently than its EC predictions, and what its explicit output is.

To address the reviewer’s azoreductase question specifically, there is currently a three-group classification scheme for these enzymes that does not neatly cover all known bacterial azoreductases (Misal and Gawai, 2018; Ryan, 2017; Zou et al., 2020). From our positive control searches of azoreductases: olsalazine and prontosil SIMMER predictions were all most-sequence similar to Class I enzymes. SIMMER nicardipine reduction predictions, however, were most sequence-similar to ArsH, an azoreductase that lies outside the three-class system (Ryan et al., 2014).

3. How did MetaCyc filter from its ~18K reactions to the nearly ~9K employed?

Good question. These numbers are clarified in our Methods section lines 811–814 and summarized here:

“We employed the Reactions dataset from MetaCyc version 24.1 which contains N=16,810 reactions. Of these reactions, only N=13,387 were enzyme-driven reactions that possessed UniProt and/or Entrez sequence linkouts. Of these, only 8,914 were fingerprintable. Failed fingerprints were due to ambiguous SMILES identifiers or presence of non-small molecule compounds in a reaction, such as peptides.”

4. What are the limitations of MetaCyc? Even 9K sounds like a lot, but given ~10 million unique proteins in a conservative estimation of the gut microbiome, those factors (even if only 10% are enzymes) are likely to catalyze more than 10,000 reactions.

MetaCyc and KEGG are currently the most informative literature-curated pathway databases, both containing more bacterial reactions than BRENDA, Reactome, or Rhea (Altman et al., 2013). In terms of numbers of single-step reactions, MetaCyc (version 24.1) exceeds KEGG (Release 104.0) (16,810 reactions versus KEGG’s 11,841). From a protein perspective too, MetaCyc exceeds KEGG in reaction representation (237,506 bacterial enzymes to KEGG’s 10,249 bacterial enzymes). Interestingly, the overlap between MetaCyc and KEGG is not as high as one would assume and was last calculated as only 1,961 reactions in common (Altman et al., 2013). Combining both bacterial chemistry databases, therefore, would be the most comprehensive representation of bacterial chemistry. For simplicity of standardized data preparation, we chose to only employ one database, MetaCyc, but SIMMER could easily be extended in the future to include more underlying databases. To satisfy the reviewer request for a comparison of chemical resources, we computed fingerprints for all KEGG reactions and plotted PCs 1-3 after performing PCA jointly on all MetaCyc and KEGG chemical reactions. The two databases largely overlap, but there does seem to be some additional chemical diversity in KEGG that SIMMER could benefit from in future iterations:

**Author response image 1. sa2fig1:** 

This is an important limitation to make the reader aware of, and we have edited our Discussion (lines 771–776) describing the limitations of MetaCyc accordingly:“While MetaCyc is the largest annotated bacterial reaction database to date (Altman et al., 2013; Caspi et al., 2020), it likely only represents a fraction of the still underexplored microbiome catalyzed biotransformation space. Additionally, the chemistry present in MetaCyc is not restricted to anaerobic organisms such as the bacteria found in the human gut. As MetaCyc expands, or additional databases get employed, SIMMER will be able to make increasingly fine-tuned predictions.”

5. The statement on lines 214-5 that the data in Figure 3B that "there is much variation in this relationship" appears to be a massive understatement. Figure 3B appears to contain much more information than the authors are considering. To name just one, what is the nature of the cluster between 0.2-0.4 on the x-axis and ~0.2 on the y-axis?

Great clarifying question. This linear relationship is not part of SIMMER; it’s a description of the relationship between proteins and chemistry in MetaCyc. This highly varied relationship is why we use pHMM searches that don’t require high global (i.e., full sequence) percent identities. Figure 3B and its significance for SIMMER’s architecture are now described more clearly in our Results section (lines 217–229):

“We also found a negative association between chemical reaction distance and global sequence similarity of MetaCyc enzyme annotations, though there is much variation in this relationship (Figure 3B). This reflects the known association between sequence similarity and similarity in chemical function, as well as reports that this relationship can often be overestimated (Tian and Skolnick, 2003). Indeed, MetaCyc reactions with identical chemical representations could be conducted by proteins with only ~20% global percent identity all the way to 100% (Figure 3B). This relationship informed our decision to use pHMM searches rather than BLAST to find sequence similar proteins from the human gut microbiome, as pHMM searches rely on matches of conserved sites rather than global percent identity between sequences. Together, these analyses informed the manner in which we combined chemical and protein similarity to create a microbiome enzyme prediction tool.”

6. On line 281, an output from SIMMER is stated to provide a list of the "top 20" ranked MetaCyc reactions. Is this good enough? The manuscript claims that this tool will provide information that will allow researchers to focus their hypotheses, but if the "right answer" is only 5% of the "possible answers" provided, that is pretty unfocused and does not support the (highly, highly) repeated claims of clean deliverables proffered by the authors.

We agree. We have edited the manuscript to describe why we chose the reaction cutoff we did for accuracy reporting in lines 283–286. Additionally, we provide an assessment of how SIMMER’s accuracy and false positive rate change as more reactions are used in the output (Figure 5—figure supplement 2). Finally, we modified SIMMER’s command-line program to make the number of reactions considered into a user-defined parameter.

7. There appears to be package version conflicts for the majority of the conda dependencies specified in SIMMER.yml, such that implementing command-line SIMMER is not straightforward as described in the manuscript or on the github page. This needs to be corrected and β-tested so users can successfully use the tool locally in a command-line manner.

We apologize for this issue, and thank Reviewers 1 and 2 for bringing it to our attention. The GitHub repository has been updated with a yml file that has been tested by independent users, along with instructions for alternatively installing required packages directly from Anaconda.

8. An additional "real world" example should be explored to examine the success of this tool. Two suggestions are PMID 31663730 or the very recently published PMID 35953888. In both cases, cleavage of glucuronides is examined from human drugs. However, very unique and hard-to-predict types of glucuronidases are identified in each case, and these unique orthologs are due to the distinct linkage of the glucuronide to the parent drug molecule. Would SIMMER be able to identify these orthologs?

We agree that evaluating SIMMER on external datasets (as we did previously with PMID32526207 and PMID31158845) is valuable. As mentioned in our original submission, SIMMER is meant to be a hypothesis-building tool for preliminary experiments on a broad set of enzymes and species, not a fine-tuned predictor of the specific residues and structures that allow a reaction to proceed. That being said, we did evaluate both of the recommnended publications and added both examples to our list of positive controls (PMID 31663730 and PMID 35953888).

The first study, PMID31663730, did not collect stool samples from a patient population, so we were only able to evaluate regorafenib-glucuronide deglucuronidation as an additional positive control reaction. Table 1, Figure 5, Supplementary Files 1-2, and the section “SIMMER captures known gut bacterial enzymes involved in drug metabolism” have been updated accordingly. To describe fully here, PMID31663730 showed that only four (CBK98066.1, WP_118096903.1, WP_118581144.1, CBJ55484.1) out of 31 tested β-glucuronidases were able to conduct reactivation of regorafenib. We evaluated which out of the 31 sequences SIMMER predicted and found that SIMMER significantly predicted the true metabolizers out of the 31 sequences (p-value=0.02, Fisher’s exact test). In addition to predicting all four of the true positives, SIMMER additionally predicted eight false positives,

WP_102727023.1 DUF4982 domain-containing protein [Akkermansia muciniphila]WP_130053801.1 malectin domain-containing carbohydrate-binding protein [Phocaeicola dorei]WP_005822521.1 DUF4982 domain-containing protein [Bacteroides]WP_022502167.1 β-glucuronidase [Lachnospira pectinoschiza]WP_001551153.1 β-glucuronidase [*Enterobacteriaceae*]HBH98519.1 β-glucuronidase [Lachnospiraceae bacterium]WP_010800018.1 β-galactosidase [Parabacteroides]CUO31707.1 Β-glucuronidase [Roseburia hominis],

resulting in a 30% enzyme prediction false positive rate for this reaction. This was a helpful study for us to explore; most pharmacomicrobiomics findings only provide true-positives, but the current study allowed us to compute a false positive rate, which was fairly high (30%) for the reaction at hand. This is in line with our expectation that while SIMMER has high accuracy, it will suffer from false positives. From the perspective of hypothesis generation for future experiments, this false positive rate is tolerable.

Another point of interest from this study is that regorafenib-glucuronide is N-linked, unlike the O-linked β-glucuronidase reactions we previously assessed as positive controls. Despite there being no MetaCyc N-glucuronide hydrolysis reactions (only O-glucuronides), SIMMER’s precomputed chemical embeddings allowed the retrieval of N-glucuronide acting β-glucuronidases. This finding interested us, so we proceeded to assess the type of structures present in SIMMER’s regorafenib reactivation predictions. Previous research shows that β-glucuronidases come in six structural flavors based on type and/or presence of an active site loop: Loop1 (L1), mini-Loop1 (mL1), Loop2 (L2), mini-Loop2 (mL2), mini-Loop1,2 (mL1,2), and No Loop (NL) (Pollet et al., 2017). All four true positives from PMID31663730 were NL β-glucuronidases (Ervin et al., 2019), and SIMMER’s predictions were from the NL (76% of SIMMER predictions) and mL1 (24% of SIMMER predictions) active site groups. This analysis demonstrated SIMMER’s ability to make relevant enzyme predictions even for reactions not explicitly described in the MetaCyc database.

The second study (PMID35953888) could have been used as an additional external dataset validation, as it included an assessment of mycophenolic acid reactivation in patient stool samples. Unfortunately, we have not yet been granted approval to download the metagenomic sequencing data from this study (a request was submitted shortly after receiving these reviews). To satisfy the request for an additional “real world example,” we therefore turned to PMID33314800 and assessed whether or not SIMMER’s enzyme predictions for Methotrexate (MTX) hydrolysis to DAMPA and glutamate were enriched in rheumatoid arthritis (RA) patients clinically unresponsive to the drug. This work is described on lines 507– 530 of the resubmitted manuscript and in the Essential revisions response 1.1.

Reviewer #2 (Recommendations for the authors):A few suggestions to improve and clarify the manuscripts.1) Presenting how many false positive hits per prediction are generated by SIMMER, belonging to how many EC classes, sub-classes, etc. together with the truly positive results would benefit the interpretation of results. Also, the rank position of the true positive prediction in the prediction list should be reported, and not only its presence, since only the top enzyme annotations are used as queries for the protein similarity search of potential gut bacterial enzymes. Especially the potential impact of the presence of several EC classes (eventually biasing the hypothesis generation process before experimental validation) should be assessed.

We have now added a false positive assessment. To answer the further questions here, we did previously report the rank position of positive controls in the Figure 5—source data file. Also, we realized after reading all reviewer responses, including this, that we did not adequately describe how protein similarity searches were conducted and that these are distinct from our predictions of EC codes. Please refer to Reviewer 1 point 2.

We agree that our analysis would benefit from an assessment of false positives. Unfortunately, current literature usually reports which reactions an enzyme is capable, rather than incapable, of performing. For this reason, we took a conservative approach and decided to define all reactions preceding that which yielded a positive control enzyme sequence as false positives. This is now described above in Essential revisions response 1.3.

2) Due to the different input formats between SIMMER, MicrobeFDT, and DrugBug, it would be useful to demonstrate that SIMMER has at least comparable performances to the other tools when their input format is used (i.e., substrate fingerprint only).

We have addressed this concern in Essential revisions response 3.

3) A revision of the text is needed to clarify terms and avoid the readers' confusion. Already in the title, the method is described as being able to "identify chemical transformations in the human gut microbiome"; however, SIMMER requires a fully chemically characterized biotransformation (in terms of substrate, (co-factors), and products) as input (and is therefore not able to identify new biotransformations). The text should be modified to highlight that SIMMER identifies and characterizes the enzymatic reaction of known biotransformation. On the same note, please also revise lines 81-82, 358-359, 480-481, and 531-532 accordingly.

We agree, and have reworded key phrases in the text, including the title.

New Title:

SIMMER employs similarity algorithms to accurately identify human gut microbiome species and enzymes capable of known chemical transformations

Former lines 81–82 (current lines 89–91) changed to:

“To address these gaps in accurate predictive software, we present SIMMER, a tool that combines chemoinformatics and metagenomics approaches to accurately predict bacterial species and enzymes capable of known biotransformations in the human gut.”

Former lines 358–359 (current lines 386–389) changed to:

“Altogether, these findings illustrate SIMMER’s enhanced accuracy over other methods for the use case of characterized drug metabolism events by gut bacteria, and also illustrate SIMMER’s novel ability to predict species and enzymes responsible for chemical transformations not previously described in literature or databases.”

Former lines 480–481 (current lines 620–623) changed to:

“This advances our ability to discover the genetic determinants of human microbiome chemical transformations, because previous methods for in silico metabolism prediction had several key limitations, including low accuracy.”

Former lines 531–532 (current lines 694–696) changed to:

“Due to its accurate predictions of previously described drug-metabolizing bacterial enzymes, we also used SIMMER to identify previously unknown bacterial species and enzymes at play in known gut microbiome drug transformations.”

4) The website version of SIMMER could be easily tested; it's intuitive, although a more extensive description of Output headers could help the user understand the results better. It was not possible to create the conda env after cloning the github repo ("Solving environment: failed" error)

We apologize for the conda environment issue, and thank both Reviewer 1 and 2 for bringing it to our attention. The GitHub repository has been updated with yml file that was tested by independent users, along with instructions for how to proceed in the case of solving environment issues.

5) As showcased in Figure4—figure supplement 1, additional filtering steps might be needed to filter the SIMMER enzyme list to a relevant subset (e.g., before in vitro validation). The authors should consider adding a plug-in to both the command line and online version of the tool so that these filtering steps could potentially be applied by the users as well. This might benefit especially experimentalist users, who might not be comfortable with running several additional filtering steps via cli.

We contacted the creators and maintainers of the Similarity Ensemble Approach (SEA) tool to see if this was possible. At this time unfortunately, the creators wish for the repository (and therefore commandline tool) to remain private. SIMMER users wishing to use SEA can use the public webtool, or likely gain access to the commandline SEA by directly contacting the creator (*Keiser M, personal communication, October 24, 2022*).

6) More detailed results might improve the understanding of the reported examples. E.g., it is unclear how SIMMER could not accurately predict EC number for brivudine transformation (Line 197), but correctly predict BT_4554 as being responsible for the reaction. Reporting SIMMER output for this query as supplementary material would help the user navigate the examples better.

Please see response to Reviewer 1, comment 2.

7) In the example describing dexamethasone biotransformation, the authors should describe the SIMMER predicted enzyme(s) in more detail (predicted EC number, sequence, etc.) since it (they) reportedly does not correspond to the already described DesAB gene.

Thank you for catching this. We have included SIMMER’s sequence and annotation prediction output for dexamethasone metabolism in the Figure 10—source data file.

8) When quantification of SIMMER predicted enzymes is performed in metagenome samples (see for example Lines 427, 430, 545), it is unclear what association is performed: is this the quantification of only the first (i.e., top scoring) predicted enzyme in the sample or is it the quantification of several different SIMMER enzymes in the same sample? If the latter, how are quantification results from different enzymes collapsed together?

The latter association is performed, and we have reworded the Materials and methods in lines 1012–1016 to make this clear:

“The metabolism of dexamethasone to 17-oxodexamethasone was input to SIMMER with N=20 reactions output, and a DIAMOND database of all SIMMER enzyme predictions from C. scindens was created. Presence of SIMMER enzyme predictions was established via search with DIAMOND and normalized by sample read depth.”

Reviewer #3 (Recommendations for the authors):I am most concerned about major overstatements related to the performance and validation of SIMMER resource predictions."We show that SIMMER predicts the chemistry and responsible species and enzymes for a queried reaction with high accuracy."

Thanks for pointing this out. We agree and have edited wording throughout to correct any overstatements. We also have added many additional validations to this manuscript version to further support our findings. These additional validations include an in vitro experimental validation of SIMMER methotrexate metabolism predictions, additional positive controls, a false positive rate, additional comparisons to competing methods, and an additional validation on an external dataset.

The validation of SIMMER predicted enzyme-drug interactions utilizes metagenomic data from a single donor stool sample in which drug metabolism was demonstrated. Other samples used as 'validation' have only 16S rRNA gene sequencing.

No—we previously evaluated SIMMER on two external datasets, PMID32526207 which includes a 16S study, and PMID31158845 which includes a *shotgun sequencing study* of 28 dexamethasone-treated patient stool samples. Reviewer 3 misread the dexamethasone metagenomics analysis as another 16S study. This is clear from our Methods section, but we failed to describe this well enough in Results. We have reworded the Results section (lines 534–547) to make it clear that the dexamethasone treated cohort was a shotgun metagenomics sequencing study:

“Dexamethasone was recently shown to be metabolized solely by Clostridium scindens (ATCC 35704) out of a collection of 76 isolates representative of the human gut microbiome (Zimmermann et al., 2019b). When the authors assessed dexamethasone metabolism in 28 shotgun sequenced human stool samples, metabolite formation varied substantially by individual, but could not be explained by C. scindens species abundance. To explore this lack of correlation in light of our findings, we assessed the abundance of C. scindens SIMMER enzyme predictions (Figure 10—source data) within each of the 28 samples (i.e., the number of C. scindens SIMMER enzyme predictions aligned to each sample’s shotgun sequencing reads) rather than each sample’s C. scindens species relative abundance. We found a significant association between metabolite formation and number of SIMMER enzyme aligned reads, and also a significant association between parent compound consumption and number of SIMMER enzyme aligned reads (Figure 10B). This underscores the importance of identifying an enzyme and quantifying its presence in metagenomic data.”

We hope that the Results section reads more clearly now and that readers will understand that validation in a multi-person shotgun metagenomics dataset was performed. Furthermore, we have included an additional shotgun sequencing study validation (Rheumatoid Arthritis patients’ clinical response to methotrexate) described in Essential Revisions 1.

As to the merits of including a validation on the 16S dataset: SIMMER predicts both species and enzymes for xenobiotics degradation. We completely agree that 16S data is not enough to confirm the relevant enzymes at play in a drug degradation event, but it is a helpful analysis to include for the demonstration of species metabolism prediction. We now state the purpose of this analysis in the manuscript (lines 577–585).

Their analysis, identifying microbial taxa via 16S in fecal samples that demonstrated metabolism of a drug does not support the claim that SIMMER "predicts the chemistry and responsible species and enzymes". There is no experimental or computational enzymatic validation; rather the authors infer from taxonomic representation in 16S data that the enzymes they predict metabolize the drug must be present in the bacteria found in the drug-metabolizing fecal samples. This validation is several steps removed from identifying "responsible species and enzymes". This sentence in the abstract is thus a major overstatement of their results.

Please see response above.

"Bacterial species containing these enzymes are enriched within human donor stool samples that metabolize the query compound."To discuss this point further; that bacterial species purported to contain specific enzymes are enriched in stool samples that metabolize a compound is insufficient to validate that SIMMER has identified microbiome enzymes that metabolize a drug. Saying that you see species increase in abundance that you think have enzymes that can metabolize a drug is far removed from their claim that they can identify specific microbiome enzyme-drug interactions.

Please see response above.

In a later discussion of microbiome metabolism of dexamethasone, the authors state: "These results indicate that species level information alone is not enough to predict chemical transformations in a microbiome sample, but with SIMMER, knowledge of responsible enzymes can recapitulate a sample's potential for therapeutic degradation." However, using species-level information alone is precisely what they did in their 16S-based validation set. This point needs to be clarified.

Please see response above.

Finally, they refer to these entirely computational validation approaches as "experimental validation" of SIMMER, which is inaccurate. Based on eLife precedent, I strongly advise experimental validation of at least one of their predicted novel enzyme-drug interactions.

We have addressed this in Essential Revisions Response 2.

Regarding sequence similarity/chemical similarity, a foundational component of the resource:"SIMMER was created with the assumption that chemically similar reactions are mediated by sequence similar enzymes"It would be helpful for the authors to consider the work of the Babbitt, Gerlt, Almo, and Jacobsen labs:https://www.ncbi.nlm.nih.gov/pmc/articles/PMC3249080/"From a survey of structurally characterized superfamilies, almost 40% are functionally diverse, i.e. different members catalyze reactions with different EC numbers (4). Thus, trivial annotation transfer by sequence homology is often not sufficient to assign function."The authors' assertion that enzyme sequence similarity is sufficient to identify chemically similar reactions is also not well supported by experimental studies on enzyme families that metabolize drugs in the human gut.In studies of Cgr2, a member of a microbiome drug-metabolizing enzyme family that was deeply characterized by the Turnbaugh and Balskus labs, sequence-based clustering was unable to resolve biochemical functioning.https://elifesciences.org/articles/33953"At all thresholds at which Cgr2 remained connected to other protein sequences, all characterized enzymes within the SSN were co-clustered, precluding the resolution of unique biochemical functions at this cutoff. "

We agree with this important caveat. It is worth noting that the works cited are assessments of global percent identity and shared function. For SIMMER we chose to use pHMM searches for this very reason. pHMM searches allow for retrieval of conserved sites in a sequence (like active and binding sites), rather than reliance on blanket global similarity, which is insufficient for functional transfer as pointed out by the Reviewer. Nonetheless, we agree completely that sequence homology, including that identified by pHMM methods, can be uncorrelated with functional similarity. To address this comment, we have added discussion of this important caveat to the manuscript with citations to the recommended papers, emphasizing that the goal of SIMMER is to prioritize and limit the set of possible experiments one might do rather than to replace experiments. SIMMER predictions will include many false positives and require experimental validation. But we believe that SIMMER nonetheless provides important information for designing and prioritizing downstream experiments.

In Figure 3B one can see the issues with protein percent identity as a metric for functional similarity. The use of a linear model is likely not appropriate here given the substantial variability in the relationship between reaction distance and percent identity. However, it is clear that the statement "reactions with similar chemistry are conducted by sequence similar enzymes" is not upheld in much of their data. As an example, in the range of identity that they are using (27% identity, per the methods) there is a large spread in reaction distances.

Please see response to Reviewer 1, comment 5.

Also, Reviewer 3 appears to misunderstand the 27% identity cutoff in the Methods. We apologize for not explaining this clearly. To create our sequence searches of microbiome gene catalogs, we choose all enzymes known to be responsible for a given reaction in MetaCyc (i.e., all enzymes in question have established *identical function*), and as long as all these *functionally identical* enzyme sequences share at least 27% sequence identity to one another, we create profile Hidden Markov Models (pHMM) to retrieve microbiome homologs. 27% identity for input to a multiple sequence alignment/pHMM is an established guideline for HMM algorithms, and it is employed widely by protein database resources like PANTHER and Pfam. We do not use 27% identity as a threshold for the discovered matches to the pHMM.

These nuances in sequence-function relationships that can confound microbiome drug metabolism studies are critical to deal with in such analyses and they are one reason that other resources that seek to identify microbiome-drug interactions utilize a variety of sources of evidence to assess the likelihood of specific microbiome-drug interactions.

We completely agree that the nuances of sequence-function relationships are critical, and for this reason took care when deciding how to retrieve sequence similar proteins that could still retain chemical function. We believe that the text reads more clearly now on this front, as our rationale for choosing pHMM search algorithms is better stated and validated. Additionally, we have updated the text to:

Discuss the limitations of linking sequence similarity to functional similarity, and,Highlight that other resources (MicrobeFDT) utilize a multi-pronged approach to identity shared function (Table 1).

As the authors note, each of these current tools to predict microbiome-drug interactions has limitations. For SIMMER, the major limitation, which is only briefly alluded to, is the need for a user to already know both substrate and product. As the authors note, publications experimentally characterizing microbiome-drug interactions often utilize parent compound loss as the marker for microbiome drug metabolism.Knowing and providing the substrates and products requires chemical knowledge that could also be used to identify the type of reaction that is being carried out. This makes the SIMMER resource harder to use for users without that chemical knowledge, which is where the other resources that the authors use as a comparison (BugDrug, MicrobeFDT) may be more helpful.

We agree that this is a limitation for the user, but as we show in our Results, relying on substrates alone does not yield appropriate representations of *reactions* and therefore does not allow for accurate predictions of responsible species/strains and enzymes (i.e., finding True Positives, and confirming associations from previously collected data). We agree that tools requiring only substrates are convenient, but our results show that they are less helpful in finding appropriate metabolism and enzyme predictions. Many studies of biotransformation in the human gut identify the product information or product structure via HPLC, LC-MS, and NMR techniques. In cases where such data was not gathered, or not gathered with enough structural resolution, researchers can use tools such as Biotransformer to make product template predictions before inputting a query to SIMMER. This recommendation is included in the present manuscript’s lines 369–384:

“In instances when DrugBug and MicrobeFDT did make predictions, they suffered from low accuracy (Table 1), which we hypothesized was due to both methods’ reliance on substrate rather than reaction chemistry. Biotransformations involve the relationship between substrate(s), cofactor(s), and an enzyme to yield a particular product(s). As one substrate can exhibit affinity for multiple enzymes, resulting in multiple unique products, sole employment of substrates in a chemical fingerprint does not achieve the resolution necessary to make relevant predictions. To test if SIMMER’s better performance could be attributed to including cofactors and products, we modified our code to run with a chemical representation that includes only the substrate of each positive control reaction. Enzyme prediction accuracy dropped from 88% down to 33%, and EC prediction accuracy dropped from 93% down to 48% (Table 1—source data), supporting the hypothesis that SIMMER’s better performance when compared to DrugBug and MicrobeFDT is due in large part to our using chemical representations that include the full reaction. These results are in line with our previous demonstration that SIMMER clusters enzymatic reaction chemistry only when a full reaction is employed (Figure 2, Figure 2—figure supplement 4).”

Additional commentsThe authors should indicate how many different modifications are represented in the 88 reactions that were queried to evaluate how well the resource works at predicting novel drug-metabolizing enzymes.

The 88 reactions in Figure 6 represent 88 distinct modifications, mostly hydrolysis events, followed by oxidoreduction and transferase events. To assess how well SIMMER predicts all 88 previously uncharacterized reactions, 88 experimental assessments would need to be performed. As this was intractable for the amount of time we had, we have now added an in vitro assessment of one of these reactions, methotrexate hydrolysis. For an additional seven reactions (dexamethasone, capecitabine, misoprostol, nicardipine, risperidone, spironolactone, and tolcapone), we performed validations using external datasets.

It is unclear if the SIMMER resource covers greater chemical space than the other resources. Database size does not necessarily correlate with chemical diversity. An explicit comparison of chemical space should be quantified and compared across all resources.

Please see our response to Reviewer 1, comment 4.

The authors do not represent the goals of the other resources that they use as comparisons accurately; these resources take different approaches to identify microbiome-drug predictions. As an example, the "Direct query" of the MicrobeFDT resource yielded 3/4 "correct" predictions, why does Table 1 not include this accuracy metric?

This is a good point, and we have updated Table 1 and the text to describe additional goals of the tools (e.g., MicrobeFDT’s network descriptions of existing data). We have updated the text, Table 1, and supplemental data files to make explicit the accuracy measures of each tool. MicrobeFDT has 29% accuracy on EC predictions (Table 1) because it correctly predicted EC classes for 4/14 substrate degradation events. To provide even more clarity:

Out of the 33 positive control reactions, MicrobeFDT only makes predictions for 14. MicrobeFDT can be queried in two ways: first by looking for direct enzyme metabolism events, and second, by looking for enzyme metabolism of compounds that overlap chemically with the positive control in question. In theory, the first manner of query should always result in 100% accuracy because it only reflects already known data. Four of the 14 reactions could be assessed in the first manner (3 of which MicrobeFDT predicted correctly), and ten reactions in the second manner (0 of which MicrobeFDT predicted correctly). This resulted in a total accuracy of 4/14, or 29%.

It would be helpful for the authors to discuss the limitations of MetaCyc and the representation of enzymes involves in anaerobic degradation reactions that would be expected in the gut.

Please see our response to Reviewer 1, comment 4.

The authors need to make available and searchable their data connecting microbiome taxa, enzymes and predicted drug metabolism, this would greatly broaden the utility of the resource to the community.

Thank you for this good suggestion. All the underlying chemical, microbiome taxa, and microbiome protein data of SIMMER is already available (via link on SIMMER’s webtool page and Github page). This data link will be updated each year in tandem with major MetaCyc updates. To make this information more searchable as you suggest though, we have made a new program within the SIMMER command line tool (metacyc_microbiome_homologs.py) that allows the user to directly look up the microbiome homologs associated with any given MetaCyc reaction. SIMMER was not created with the intention of providing a searchable network of all microbiome proteins and chemistry, but for the curious researcher this would be doable with this newly provided program.

Artacho A, Isaac S, Nayak R, Flor-Duro A, Alexander M, Koo I, Manasson J, Smith PB, Rosenthal P, Homsi Y, Gulko P, Pons J, Puchades-Carrasco L, Izmirly P, Patterson A, Abramson SB, Pineda-Lucena A, Turnbaugh PJ, Ubeda C, Scher JU. 2020. The Pre-treatment Gut Microbiome is Associated with Lack of Response to Methotrexate in New Onset Rheumatoid Arthritis. *Arthritis Rheumatol*. doi:10.1002/art.41622

Ervin SM, Hanley RP, Lim L, Walton WG, Pearce KH, Bhatt AP, James LI, Redinbo MR. 2019. Targeting Regorafenib-Induced Toxicity through Inhibition of Gut Microbial β-Glucuronidases. ACS Chem Biol 14:2737–2744. doi:10.1021/acschembio.9b00663 Misal SA, Gawai KR. 2018. Azoreductase: a key player of xenobiotic metabolism. Bioresources and Bioprocessing 5:17. doi:10.1186/s40643-018-0206-8

Pollet RM, D’Agostino EH, Walton WG, Xu Y, Little MS, Biernat KA, Pellock SJ, Patterson LM, Creekmore BC, Isenberg HN, Bahethi RR, Bhatt AP, Liu J, Gharaibeh RZ, Redinbo MR. 2017. An Atlas of β-Glucuronidases in the Human Intestinal Microbiome. Structure 25:967–977.e5. doi:10.1016/j.str.2017.05.003

Ryan A. 2017. Azoreductases in drug metabolism. Br J Pharmacol 174:2161–2173. doi:10.1111/bph.13571

Ryan A, Kaplan E, Nebel J-C, Polycarpou E, Crescente V, Lowe E, Preston GM, Sim E. 2014. Identification of NAD(P)H quinone oxidoreductase activity in azoreductases from *P. aeruginosa*: azoreductases and NAD(P)H quinone oxidoreductases belong to the same FMN-dependent superfamily of enzymes. PLoS One 9:e98551. doi:10.1371/journal.pone.0098551

Zou L, Spanogiannopoulos P, Pieper LM, Chien H-C, Cai W, Khuri N, Pottel J, Vora B, Ni Z, Tsakalozou E, Zhang W, Shoichet BK, Giacomini KM, Turnbaugh PJ. 2020. Bacterial metabolism rescues the inhibition of intestinal drug absorption by food and drug additives. Proc Natl Acad Sci U S A 117:16009–16018. doi:10.1073/pnas.1920483117

Altman T, Travers M, Kothari A, Caspi R, Karp PD. 2013. A systematic comparison of the MetaCyc and KEGG pathway databases. *BMC Bioinformatics* 14:112. doi:10.1186/1471-2105-14-112

Caspi R, Billington R, Keseler IM, Kothari A, Krummenacker M, Midford PE, Ong WK, Paley S, Subhraveti P, Karp PD. 2020. The MetaCyc database of metabolic pathways and enzymes – a 2019 update. *Nucleic Acids Res* 48:D445–D453. doi:10.1093/nar/gkz862

Pouliot Y, Karp PD. 2007. A survey of orphan enzyme activities. *BMC Bioinformatics* 8:244. doi:10.1186/1471-2105-8-244

Zimmermann M, Zimmermann-Kogadeeva M, Wegmann R, Goodman AL. 2019. Mapping human microbiome drug metabolism by gut bacteria and their genes. *Nature*. doi:10.1038/s41586-019-1291-3

[Editors' note: further revisions were suggested prior to acceptance, as described below.]

The manuscript has been improved but there are some remaining issues that need to be addressed, as outlined below.Reviewer #3 (Recommendations for the authors):I am satisfied that the authors have carried out key validations of the SIMMER tool, including experimental characterization of a SIMMER prediction, the metabolism of methotrexate by hydrolases.I ask that the authors credit prior research demonstrating that methotrexate is metabolized by hydrolases (https://www.ncbi.nlm.nih.gov/pmc/articles/PMC5082436/). They extend prior work on methotrexate here by showing strain-level characterization of MTX metabolism, a valuable addition to our understanding of drug metabolism by gut microbes.

Thank you for this suggestion. We have now added this and another relevant citation to lines 458-461 of the current manuscript:

“When queried with MTX and its gut bacteria associated metabolites DAMPA and glutamate, SIMMER calculated a most similar MetaCyc reaction (3.4.17.11-RXN) and a significant EC prediction (3.4.17.11, p-value<0.001). This MetaCyc reaction describes the hydrolysis of folate into pteroate and glutamate, driven by a glutamate carboxypeptidase (Cpg2) found in environmental *Pseudomonas aeruginosa*. Hydrolysis of MTX is chemically similar to hydrolysis of folate (Figure 7A) with a Tanimoto coefficient=0.6, and normalized euclidean distance=0.05 in SIMMER’s precomputed chemical space. SIMMER made 2,286 human gut microbiome enzyme predictions for degrading MTX into DAMPA and glutamate (Figure 7—source data). The most common Prokka annotation for these enzymes is Carboxypeptidase G2s (Figure 7B) due to their sequence similarity to MetaCyc’s environmental Cpg2. Furthermore, SIMMER’s predicted enzymes had a median global identity of 33% to *Pseudomonas sp* RS^-1^6 (an environmental bacterium) Cpg2, an enzyme known to conduct hydrolysis of MTX (Larimer et al., 2014; Jeyaharan et al., 2016).”